# FtsZ phosphorylation modulates tail-core binding to tune cell division in *Bacillus subtilis*

William J. Mallard*, Vincent V. Pham*

Department of Molecular and Cellular Biology, Harvard University, Cambridge, Massachusetts, United States of America

* wmallard@fas.harvard.edu (WM); vpham@g.harvard.edu (VP)

## Abstract

The cell division protein FtsZ contains an intrinsically disordered C-terminal tail whose function remains poorly understood. Here we demonstrate that this tail serves as an autoinhibitory element through intramolecular interaction with FtsZ's globular core. In *Bacillus subtilis*, we show that FtsZ's tail is phosphorylated at serine 333 by the kinase PrkC during vegetative growth. We establish that the tail binds specifically to the core's C-terminal polymerization surface through a molecular recognition element spanning L330-H337. Using NMR spectroscopy, we reveal that phosphorylation of S333 induces structural collapse of the I334-K335 motif, sequestering these key residues from their binding sites and disrupting the tail-core interaction. Mutations at S333 reduce the critical concentration for polymerization and enhance GTPase activity *in vitro* while altering cell length *in vivo*. Competition experiments demonstrate that tail-core binding occludes the division inhibitor MinC from the polymerization surface, with S333 mutations restoring MinC binding. Our findings reveal that FtsZ's tail gates access to the polymerization surface for other FtsZ monomers as well as regulatory proteins, with phosphorylation serving as a molecular switch that coordinates licensing of division regulators and FtsZ polymerization dynamics during the cell cycle.

## Introduction

FtsZ, the central coordinator of bacterial cell division [1–3], consists of a globular core region and a mostly intrinsically disordered C-terminal tail (CTT) [4–6] (Fig 1A, S1). FtsZ cores polymerize into filaments that assemble into a ring-like structure (the Z-ring) at the division site [7–9]. Like other cytoskeletal proteins, FtsZ only forms filaments above a threshold protein concentration (the critical concentration), below which the proteins remain largely monomeric. These filaments exhibit GTPase activity at core-core interfaces, driving dynamic turnover through GTP hydrolysis [10–15]. FtsZ's tail consists of two regions: a conserved C-terminal peptide (CTP) that binds anchor proteins FtsA [9], EzrA [16], and SepF [17] to tether the Z-ring to the cell membrane, and a disordered C-terminal linker (CTL) that connects the CTP to the

**Data availability statement:** All relevant data are within the paper and its Supporting Information files.

**Funding:** The author(s) received no specific funding for this work.

**Competing interests:** The authors have declared that no competing interests exist.

core [5]. While both the core and CTP have well-characterized roles in division, the function of the CTL remains poorly understood.

The CTL exhibits hallmarks of disordered flexible regions, including poor sequence conservation and length variation across species [5]. It is necessary for FtsZ function, as deletion of the CTL is lethal [6]. Cells tolerate moderate changes to CTL length, but extreme shortening or lengthening disrupts cell division [6,18]. Similarly, replacing the CTL with a rigid structure of comparable size also causes division defects [6,18]. These observations have led to the view that the CTL serves primarily as a flexible mechanical tether, with length and flexibility as its only constraints. This view is further supported by the observation that *B. subtilis* cells remain viable even when the CTL is randomly scrambled [6], suggesting the specific amino acid sequence of this linker is unlikely to be functionally relevant.

Despite the apparent tolerance for changes in the CTL, specific modifications to this region can significantly impact FtsZ function. In a number of bacterial species, the FtsZ CTL is phosphorylated by homologs of *B. subtilis* PrkC [19–28] (Table A in S1 Appendix), a Hanks-type Serine/Threonine kinase [29–32] known to phosphorylate other cell cycle proteins [33]. While the phosphosites vary across species due to the region's poor sequence conservation, CTL phosphorylation consistently alters FtsZ GTPase activity and polymerization dynamics. Since GTP hydrolysis occurs at core-core interfaces within filaments [14], these observations suggest the tail somehow modulates polymerization. Supporting this possibility, molecular dynamics simulations predict that the tail binds the core on its polymerization surface [34], where other cell cycle-dependent regulators of FtsZ polymerization, such as MinC, are also known to bind [35,36]. While these findings suggest interaction between the tail and core, direct physical binding has never been demonstrated, and the mechanism by which phosphorylation affects polymerization remains unclear.

Here we demonstrate that FtsZ in *Bacillus subtilis* is phosphorylated at serine 333 within the tail by the kinase PrkC during vegetative growth. Mutations at S333 promote FtsZ polymerization and enhance GTPase activity *in vitro* while producing context-dependent effects on cell length *in vivo*. We show that the tail binds directly to the core's C-terminal polymerization surface, with tail residues L330-H337 forming the binding interface. NMR analysis of the binding site reveals that S333 phosphorylation drives I334-K335 compaction, sequestering these residues from their binding partners on the core. Competition experiments demonstrate that tail-core binding occludes the division inhibitor MinC, with disruption of this binding restoring MinC access. Our results reveal an autoinhibitory mechanism where FtsZ's tail gates access to the polymerization surface for both other monomers and regulatory proteins, with S333 phosphorylation serving as a molecular switch.

## Results

### FtsZ's CTL is phosphorylated at S333 by PrkC

Previous phosphoproteomic screens identified two potential phosphosites in FtsZ's CTL: S329 and S333 [37–39]. To test if FtsZ is phosphorylated *in vivo*, we harvested the lysate from log phase cells grown in S7 minimal medium and used PhosTag

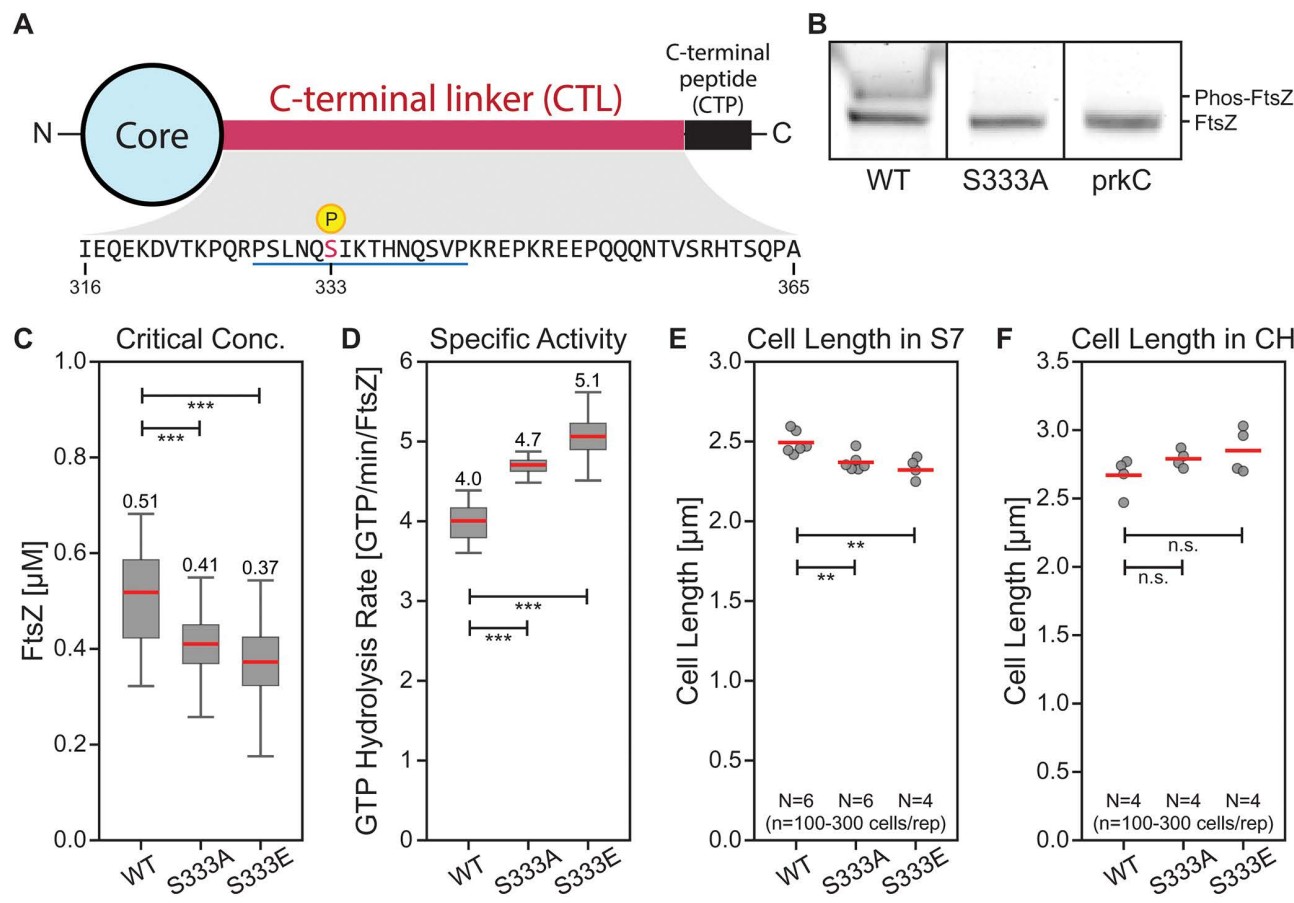

**Fig 1. Phosphorylation of FtsZ at Serine 333 regulates its activity and cell size. (A)** Schematic representation of *B. subtilis* FtsZ. The protein consists of a conserved globular core (residues 1-315, light blue), followed by an intrinsically disordered C-terminal linker (CTL, residues 316-365, magenta) containing the S333 phosphorylation site, and a conserved C-terminal peptide (CTP, residues 366-375, black) that mediates membrane teth-ering through interactions with FtsA, EzrA, and SepF. Blue underline indicates the 15aa peptide (P328-P342) used for structural studies. **(B)** PhosTag gel analysis of FtsZ phosphorylation. HaloTag-FtsZ proteins were labeled with TMR, harvested from log-phase *B. subtilis* cells grown in S7 minimal medium, and separated on PhosTag gels to resolve phosphorylated species. Wild-type FtsZ shows two distinct bands, with the upper band representing phosphorylated FtsZ. The S333A mutant eliminates the upper band, demonstrating that S333 is the sole phosphorylation site under these conditions. The Δ*prkC* strain also lacks the upper band, indicating that PrkC is required for FtsZ phosphorylation. Representative gel from n=3 (WT), n=2 (S333A), and n=2 (ΔPrkC) independent experiments. **(C)** Critical concentration of FtsZ polymerization. S333 mutations reduce the protein concentration thresh-old required for filament assembly. Wild-type: 0.51±0.17 µM; S333A: 0.41±0.12 µM (20% decrease, p<0.001); S333E: 0.37±0.14 µM (27% decrease, p<0.001). Values represent mean±95% CI from 1,000 bootstrap samples. **(D)** Specific activity of FtsZ measured above critical concentration. S333 mutations enhance the rate of GTP hydrolysis. Wild-type: 4.0±0.4 GTP/min/FtsZ; S333A: 4.7±0.2 GTP/min/FtsZ (19% increase, p<0.001); S333E: 5.1±0.4 GTP/min/FtsZ (28% increase, p<0.001). Values represent mean±95% CI from 1,000 bootstrap samples. **(E, F)** Cell length analysis of FtsZ S333 mutants. Cells expressing wild-type (bWM200) or mutant FtsZ (S333A: bWM236, S333E: bWM237) were grown to early log phase and imaged using FM 5-95 membrane stain. **(E)** In S7 minimal medium, both S333A and S333E mutations resulted in shorter cells compared to wild-type (2.37±0.06 µm and 2.32±0.07 µm vs. 2.49±0.07 µm; 5% and 7% decrease, respectively, p<0.01). **(F)** In CH rich medium, the same mutations produced longer cells (2.79±0.06 µm and 2.85±0.17 µm vs. 2.67±0.14 µm; 5% and 7% increase, respectively, p>0.05). Each point represents the median cell length from one experimental replicate (~100-300 cells per replicate). Statistical significance was assessed by Student's t-test.

gels to resolve differentially phosphorylated species of fluorescently tagged FtsZ. For wild-type FtsZ, we observed the emergence of a second upper band, indicating that FtsZ is phosphorylated at one or more sites *in vivo* (Fig 1B, S2A). An FtsZ S329A-S333A double mutant showed complete loss of the upper band, confirming phosphorylation occurs at one of these residues (S2B Fig). To identify the specific phosphosite, we created FtsZ variants with alanine substitutions at each

putative site. While the S329A mutant retained the upper phosphorylated band, the S333A mutant eliminated it (Fig 1B, S2C), demonstrating that FtsZ is singly phosphorylated at S333 under these conditions.

We next sought to identify the kinase responsible for this modification. While bacteria primarily use histidine kinases for signal transduction, some also encode a small number of Ser/Thr kinases that resemble eukaryotic kinases (so-called Hanks-type kinases) [29–32]. *B. subtilis* has four such kinases: PrkC, PrkD, YabT, and YrzF. We focused on PrkC because its homologs are known to phosphorylate FtsZ in other bacterial species (Table A in S1 Appendix) and it is expressed during vegetative growth [40]. Deletion of *prkC* eliminated the upper band (Fig 1B, S2D), indicating that this kinase is either directly or indirectly required for S333 phosphorylation *in vivo*.

### S333 mutations alter FtsZ critical concentration and specific activity

To test whether S333 mutations affect FtsZ polymerization, we substituted S333 with glutamate (S333E) or alanine (S333A). Malachite green-based GTPase assays revealed that these mutations affected both FtsZ's critical concentration and GTPase activity (S3A-D Fig). These mutations significantly reduced FtsZ's critical concentration for assembly, from $0.51 \pm 0.17$ μM for wild-type to $0.41 \pm 0.12$ μM and $0.37 \pm 0.14$ μM for S333A and S333E, respectively (20% and 27% decreases, $p < 0.001$; mean $\pm 95\%$ CI) (Fig 1C). The mutations also significantly enhanced FtsZ's GTPase activity: compared to wild-type FtsZ's rate of $4.0 \pm 0.4$ GTP/min/FtsZ, the S333A and S333E mutants showed significantly higher rates of $4.7 \pm 0.2$ and $5.1 \pm 0.4$ GTP/min/FtsZ, respectively (19% and 28% increases, $p < 0.001$; mean $\pm 95\%$ CI) (Fig 1D). Taken together, these observations show that FtsZ's polymerization and activity are sensitive to S333 mutations.

### S333 mutations affect cell length

To determine whether S333 mutations affect cellular physiology, we measured cell lengths in strains expressing a modified *ftsAZ* operon (with wild-type *ftsA* and mutant *ftsZ*) as the sole copy from an ectopic locus under the native *ftsAZ* promoter (*Paz*, [41]) (S4A-C Fig). This design approximates native expression levels and ensures comparable protein levels across mutants, which is essential for phenotypic analysis since cell size is sensitive to FtsZ concentration. Western blot analysis confirmed that FtsZ protein levels were comparable across wild-type, S333A, and S333E strains (bWM200, bWM236, and bWM237, respectively; S4D Fig).

We measured cell lengths from fluorescence micrographs of membrane-stained cells. To accurately assess the impact of S333 mutations and control for batch effects, we analyzed multiple strains on each day, with measurements repeated both within and across days. We focused our analysis on cells within chains of three or more, as isolated cells and cell pairs showed systematically different length distributions. For each strain, this approach yielded approximately 100–300 cells per experiment.

In S7 minimal medium during early log phase, both S333A (bWM236) and S333E (bWM237) mutant cells were shorter than wild-type (bWM200): $2.37 \pm 0.06$ μm and $2.32 \pm 0.07$ μm compared to $2.49 \pm 0.07$ μm, respectively (5% and 7% decrease, $p < .01$, Student's t-test; Fig 1E). The magnitude of these effects is notable, as previous studies have shown that doubling the amount of FtsZ only reduces cell length by 10% [42]. Our observation that a single mutation in the supposedly non-functional CTL can produce more than half of this effect size suggests strong involvement of the CTL in regulating cell division.

To test whether this phenotype was specific to minimal media conditions, we also measured cell lengths in CH rich medium. Surprisingly, S333 mutations produced the opposite effect in nutrient-rich conditions: S333A and S333E cells measured $2.79 \pm 0.06$ μm and $2.85 \pm 0.17$ μm compared to wild-type at $2.67 \pm 0.14$ μm (5% and 7% increase, $p < 0.17$ and $p < 0.13$, respectively; Fig 1F). This nutrient-dependent reversal of phenotype suggests that S333 phosphorylation modulates FtsZ's response to different growth conditions, potentially through differential interactions with nutrient-dependent regulators of cell division such as UgtP [43].

## FtsZ's tail binds its core *in vitro*

Recent molecular dynamics simulations predicted that FtsZ's tail might interact intramolecularly with its core, with the highest contact frequency in the region centered on S333 [34]. To test this prediction, we separately purified FtsZ's globular core and tail (residues 1–315 and 316–382, respectively) and conducted *in vitro* binding assays using isothermal titration calorimetry (ITC) [44]. ITC experiments revealed specific, high-affinity binding between the FtsZ tail and core with 1:1 stoichiometry ($K_d = 228 \pm 24$ nM, N = 1 ± 0.04; Fig 2A). Nuclear magnetic resonance (NMR) experiments provided additional clear evidence of complex formation, revealing striking changes in the [15]N-labeled tail spectrum upon addition of unlabeled core (Fig 2B). These dramatic spectral changes are consistent with the sub-micromolar binding we measured via ITC. Together, these results demonstrate that FtsZ's tail binds directly to its core with high affinity *in vitro*.

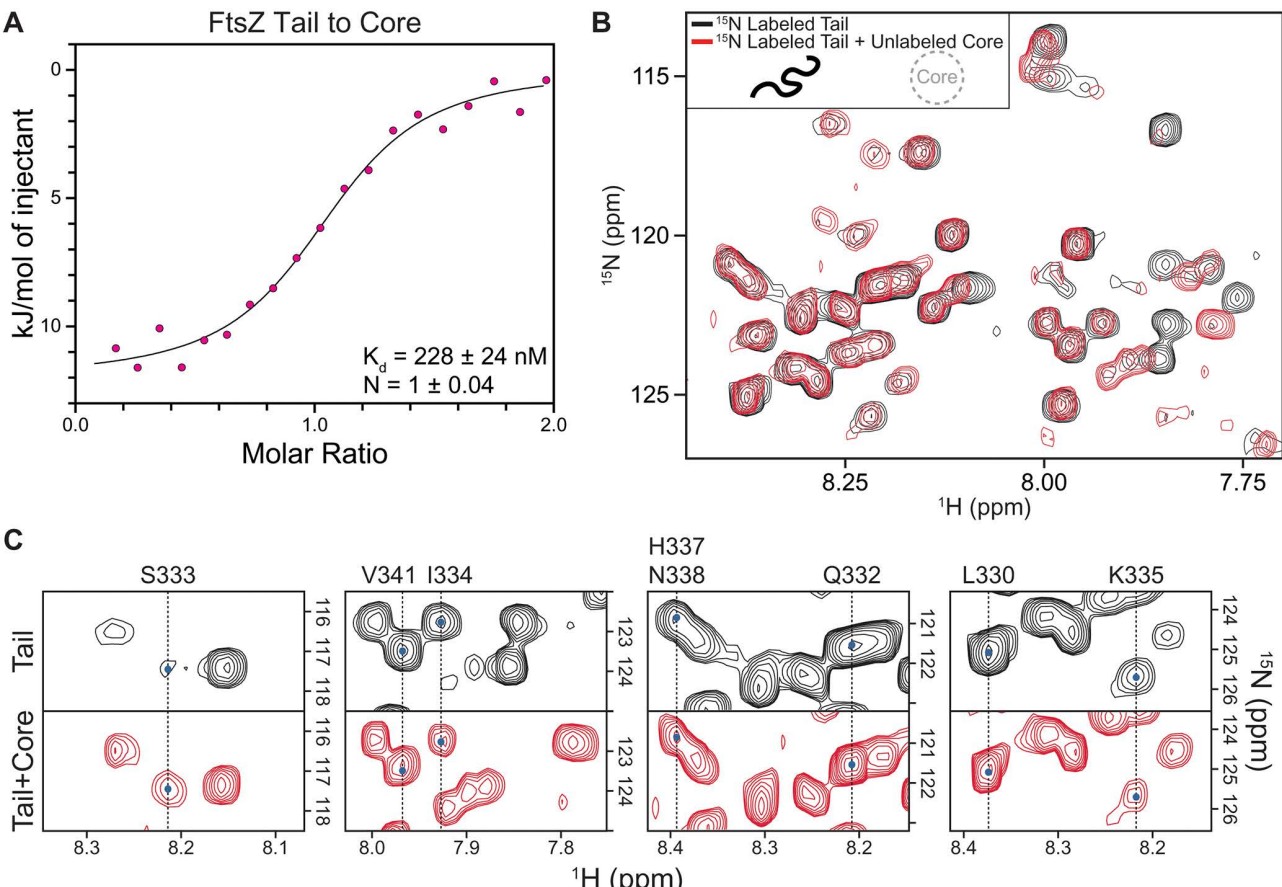

**Fig 2. The FtsZ tail binds the core via the L330-H337 binding interface. (A)** Isothermal titration calorimetry demonstrates direct binding between the FtsZ tail (residues 316-382) and core (residues 1-315). The interaction shows high affinity ($K_d = 228 \pm 24$ nM) and 1:1 stoichiometry. The continuous line represents the fit to a one-site binding model. **(B)** [1]H-[15]N HSQC NMR spectra of the 67-residue [15]N-labeled tail alone (black) and in complex with unlabeled core (red, 1:1 molar ratio). Each peak represents a backbone amide, with its position and intensity reflecting its chemical environment. Spectra are displayed at equivalent contour levels. The dramatic changes in peak positions and intensities throughout the spectrum indicate extensive tail-core interaction. **(C)** Spectral changes for assigned residues in the S333 region upon core binding. Four regions from [1]H-[15]N HSQC NMR spectra show [15]N-labeled tail alone (black) and bound to unlabeled core at 1:1 molar ratio (red). Blue dots mark the original peak positions in the free tail. Peak intensity changes reveal distinct binding behaviors: S333 shows dramatic intensity increase (141%); I334 and K335 show substantial decreases (32% and 40%); flanking residues L330, Q332, H337/N338, and V341 show modest changes (±10%). Quantification shown in S5B Fig. These response patterns indicate differential involvement of each residue in tail-core binding, with the S333-I334-K335 motif forming the binding hotspot.

## NMR identifies core binding residues on the tail

Our bioinformatic secondary structure predictions indicated α-helical propensity around S333, from L330 to H337 (Table B in S1 Appendix), in agreement with recent molecular simulations [34]. We therefore hypothesized that this region of the tail might function as an IDR molecular recognition element. To investigate this region, we synthesized a 15aa peptide spanning residues 328–342 for further NMR analysis, choosing these boundaries because proline residues typically disrupt helices. We collected $^{15}$N natural abundance $^{1}$H-$^{15}$N HSQC spectra of the 15aa peptide to observe its backbone amide peaks and used $^{1}$H-$^{1}$H NOESY to assign their identities. To identify corresponding residues in the 67aa full-length tail, we overlaid the HSQC spectra of the 15aa peptide onto the 67aa tail. Peak overlap allowed us to assign eight residues in the 67aa tail spectrum: L330, Q332, S333, I334, K335, H337/N338 (overlapped peaks), and V341 (S5A Fig, S2 Table).

All eight assigned residues showed spectral changes upon binding (Fig 2C). To quantify these changes, we compared peak intensities between the free tail and the tail-core complex (S5B Fig, S3 Table). Three contiguous residues showed the most dramatic changes: S333's peak intensity increased by 141%, while the adjacent I334 and K335 decreased by 32% and 40%, respectively. Neighboring residues showed more modest changes of around ±10% (Q332 and H337/N338 increased; L330 and V341 decreased). The striking intensity increase for S333 is consistent with this residue becoming more ordered upon binding, while the intensity decreases for I334 and K335 suggest these residues experience changes in their chemical environment, potentially through direct engagement with the core surface. Together, the S333-I334-K335 motif appears to form a binding hotspot at the center of the tail-core interface, with S333 potentially serving as a conformational switch flanked by I334 and K335 as key contact residues.

## AlphaFold identifies binding residues on the core

To identify where these tail residues bind the core, we split FtsZ into its core and CTL sequences (315aa and 50aa, respectively) and co-folded them using AlphaFold Multimer [45,46]. Although the CTL is an IDR, AlphaFold consistently predicts that its S333 region binds the core on its C-terminal polymerization surface (Fig 3A, see Methods). Furthermore, out of the entire 50aa CTL, the S333-I334-K335 motif showed the highest prediction confidence relative to the core (lowest PAE, ~12Å; S6 Fig). Remarkably, this computationally predicted S-I-K site matches the one we identified via NMR.

On the other side of the interface, AlphaFold consistently predicts that the CTL's L330-H337 region interacts with two patches on the core: (i) α-helix H10/β-strand S9, and (ii) turn T7/α-helix H8. The predicted binding interface comprises three main features: (1) a hydrophobic interface, where I334 anchors into a pocket formed by M292, I293, and F294 on S9 (the "MIF pocket"), while L330 makes additional contacts on H10/S9; (2) an ionic interface, where K335 and H337 interact with acidic residues D213 (the GTPase catalytic residue) on H8 and D210 on T7, respectively; and (3) a polar interface, where N331, Q332, and S333 interact with residues across the H10-S9 and T7-H8 patches.

Since the CTL is directly attached to the core in full-length FtsZ, any predicted binding mode must be geometrically feasible as an intramolecular interaction. Though we modeled the core and CTL as separate proteins without any constraints on position or orientation, AlphaFold nevertheless converged on a binding mode that satisfies these geometric requirements. Using an X-ray crystal structure of the FtsZ core (PDB: 2VAM [47]), we measured the distance from G314 Cα (where the CTL exits the core) to I293 Cα (in the MIF pocket) as ~14 Å. This is well within reach of the 20aa linker connecting G314 to I334, which has an average end-to-end distance of 17–23 Å as a random coil (see Methods), providing ample length for the interaction.

These structural models extend our NMR findings in two important ways. First, they identify specific residues on the core for experimental validation: the MIF pocket (M292/I293/F294) and the acidic patch around the catalytic site (D210/D213). Notably, these residues map to the core's C-terminal polymerization surface—the same surface where FtsZ monomers bind during filament assembly—suggesting that tail-core binding could regulate filament assembly by competing with monomer-monomer interactions. Second, the models reveal how phosphorylation might disrupt binding: S333 sits between hydrophobic anchors L330 and I334, and its phosphorylation would introduce a bulky negative charge that might prevent these residues from engaging the core.

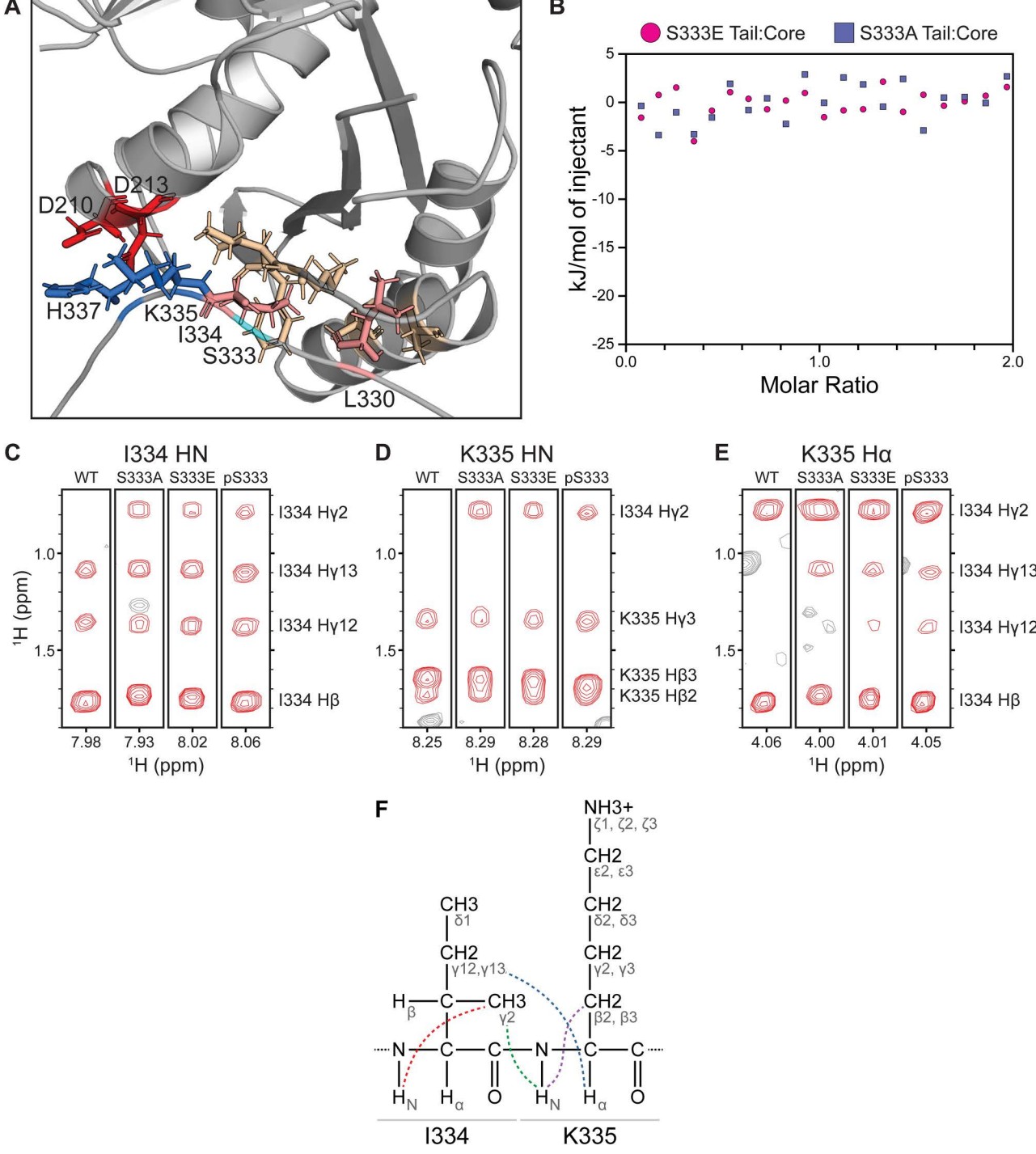

**Fig 3. Structural basis for phosphorylation-induced disruption of tail-core binding. (A)** AlphaFold model of the tail-core binding interface. Alpha-Fold Multimer predicts the tail's C-terminal linker (CTL) binds the core's C-terminal polymerization surface. Acidic residues (D210, D213) are shown in red, basic residues (K335, H337) in blue, tail hydrophobic residues (L330, I334) in salmon, and core hydrophobic residues (A279, A283, M292, I293, F294) in wheat. The S333 phosphosite is shown in cyan. **(B)** ITC demonstrates that S333 mutations abolish tail-core binding. Representative thermograms show no detectable interaction for S333E (magenta) or S333A (blue) tail variants when titrated into FtsZ core ($K_d > 2$ μM). **(C-E)** $^1$H-$^1$H NOESY spectra reveal structural changes in the tail's binding region in response to S333 modifications. Spectral strips show NOEs to **(C)** I334 HN, **(D)** K335 HN, and **(E)** K335 Hα for 15-residue peptides (P328-P342) with wild-type, S333A, S333E, or phospho-S333 sequences (left to right). **(F)** Fischer projection of the I334-K335 dipeptide showing newly appearing NOE connectivities as colored dashed lines: red (I334 Hγ2–HN intraresidue, based on panel C),

green (K335 HN–I334 Hγ2 interresidue, based on panel D), and blue (K335 Hα–I334 Hγ12/Hγ13 interresidue, based on panel E). The K335 HN–Hβ2/Hβ3 intraresidue NOEs (purple) progressively coalesce across variants (panels C-E; S9 Fig). Conserved NOEs are not shown for clarity. The modifications induce progressive structural changes following an allelic series (WT < S333A < S333E ≈ pS333), with intensification of I334-K335 NOEs and emergence of new cross-peaks demonstrating that these residues collapse together. This compaction sequesters I334 and K335 from their core-binding partners, explaining the loss of tail-core binding observed in panel B.

## Tail mutations abolish core binding

Having identified key binding residues via NMR and AlphaFold, we used ITC to test whether they are functionally required for binding. We started with the S333 phospho-site, which lies in the S-I-K motif at the center of the tail's L330-H337 binding interface. Both phospho-mimetic (S333E) and phospho-dead (S333A) substitutions reduced tail-core binding below our detection limit ($K_d > 2$ μM; Fig 3B). This represents at least a 10-fold reduction in affinity compared to wild-type ($K_d = 228$ nM), showing that the S333 position is highly sensitive to modification.

We then targeted the hydrophobic anchor, I334, which AlphaFold predicts to insert into the MIF pocket. Substituting a bulky, charged residue at this position (I334E) also reduced binding below our detection limit ($K_d > 2$ μM; S7A Fig), confirming that hydrophobic packing is essential. Finally, we tested the ionic interface by mutating the two residues that AlphaFold predicts interact with D210/D213 on the core. A K335A/H337A double mutant also showed no detectable binding ($K_d > 2$ μM; S7A Fig), validating the importance of these predicted ionic interactions. These results confirm that the S-I-K motif identified by both NMR and AlphaFold is required for tail-core binding.

## Core mutations abolish tail binding

Having demonstrated that specific tail residues are essential for binding, we next tested whether their predicted partners on the core are similarly required. We designed mutations targeting the hydrophobic and ionic interaction surfaces identified by AlphaFold. For the MIF pocket, we created a triple mutant (M292H/I293W/F294K) to introduce bulky and charged residues that would disrupt hydrophobic packing with I334. This mutation displayed aberrant binding with a biphasic isotherm (S7B Fig), confirming that the hydrophobic MIF pocket is essential for normal tail-core interaction. For the ionic interface, we neutralized the acidic patch (D210A/D213A) predicted to interact with K335/H337. This mutation weakened tail binding approximately two-fold, increasing the $K_d$ from $228 \pm 24$ nM for wild-type to $501 \pm 64$ nM (S7C Fig), confirming that the D210/D213 acidic patch also contributes to the tail-core interaction. These observations confirm that the tail binds the core's C-terminal polymerization surface, and they further pinpoint two specific structural elements of this interface: the hydrophobic MIF pocket and the D210/D213 acidic patch.

## NMR reveals the structural basis of binding disruption

To understand the structural basis of binding disruption, we examined how S333 modifications affect the local structure of the tail's binding region. We synthesized variants of our 15aa peptide (P328-P342) with S333A, S333E, and phospho-S333 modifications, and collected ¹H-¹H NOESY and ¹H-¹⁵N HSQC spectra to compare with the wild-type peptide characterized above (S8 Fig and S4–S6 Tables). Our analysis focused on understanding how modifications at position 333 affect the S-I-K motif that forms the core of the binding interface, as well as the broader L330-H337 region.

NOESY analysis revealed that S333 modifications cause the I334 and K335 sidechains to collapse and pack against each other, following a progressive series: WT < S333A < S333E ≈ pS333. Within individual sidechains, I334 showed dramatic emergence of the Hγ2-HN NOE that was essentially absent in WT but appeared strongly in all modified peptides (5-fold increase; Fig 3C), indicating the I334 γ-branch shifts from an extended conformation into closer proximity with the backbone. K335 displayed progressive coalescence of its Hβ2/Hβ3 peaks across peptide variants, with the characteristic doublet in WT merging into a single broad peak in pS333 (Fig 3D, S9), suggesting progressive restriction of the sidechain into a stable rotamer. Consistent with this, K335's Hβ2-Hγ3 NOE intensity increased progressively across variants

(S333A, +34%; S333E, +52%; pS333, +61% relative to WT; S8 Fig), indicating the β and γ methylenes move closer together as the sidechain becomes more compact. Between the two sidechains, all S333 modifications drove I334 and K335 into closer proximity: K335 HN gained new NOEs to I334 Hγ2 in all three variants (Fig 3D); K335 Hα strengthened NOEs to I334 Hγ2 (Fig 3E); and K335 Hα gained new NOEs to I334 Hγ13 (A/E/pS) and Hγ12 (E/pS only) (Fig 3E). These patterns demonstrate that S333 modifications cause the extended I334 and K335 sidechains to collapse back toward the peptide backbone and pack against each other—sequestering them from their binding partners on the core.

To determine the full spatial extent of these structural perturbations, we analyzed HSQC spectra and calculated chemical shift perturbations (CSPs) for each residue relative to wild-type (S10A Fig, S5 Table). These analyses revealed backbone conformational changes spanning residues L330-H337 with the strongest effects on the S333-I334-K335 motif (S10B Fig). S333A caused the largest backbone perturbations at I334 and moderate perturbations at K335 (CSPs of 0.46 and 0.12, respectively), consistent with substantial local conformational reorganization following loss of the serine hydroxyl. In contrast, S333E and pS333 produced smaller backbone perturbations at I334 (CSPs of 0.09 and 0.08, respectively) and K335 (CSPs of 0.08 and 0.07, respectively), suggesting these two variants produce comparable changes in backbone conformation at the binding interface.

Peak intensity changes complemented the CSP analysis by revealing the dynamic consequences of these backbone conformational changes, with distinct stabilization patterns for each variant (S10C Fig, S5 Table). S333A primarily stabilized the L330-I334 region (~30% intensity gains at L330 and I334) while destabilizing H337-S340 (~20% losses), suggesting local hydrophobic stabilization from hydroxyl loss. In contrast, S333E and pS333 stabilized the broader L330-H337 region, with moderate gains at L330 and I334 (~10–30%) and dramatic stabilization of H337 (~30–40% gains). The I334 intensity gains across all variants corroborate the conformational restriction observed via NOESY (Fig 3C).

These complementary NMR analyses reveal how each variant achieves I334-K335 compaction through distinct mechanisms. S333A forms new hydrophobic contacts between Q332, A333, and I334, as evidenced by NOEs between Q332 Hβ2, Hβ3, and Hγ2 sidechain protons and A333 HN, as well as between A333 Hβ and I334 HN (S11A-B Fig). This local hydrophobic clustering drives I334-K335 compaction while destabilizing the C-terminal region. In contrast, S333E achieves comparable I334-K335 compaction (Fig 3C-E) without novel hydrophobic NOEs, likely through electrostatic effects that stabilize the entire binding element including H337. Phospho-S333 combines features of both: Q332 sidechain repositioning similar to S333A (S11C Fig) and stabilization of the L330-H337 region (S10C Fig), suggesting that S333E provides a conservative estimate of phosphorylation's full structural impact.

Together, these NMR data demonstrate that S333 modifications disrupt the entire L330-H337 binding interface. Compaction of I334-K335 is the dominant structural change across all variants; based on our binding models and ITC results, this likely sequesters these two residues from their core partners. S333A achieves this compaction through hydrophobic contacts, while S333E and phospho-S333 achieve comparable I334-K335 compaction and stabilization of the broader L330-H337 region, likely through electrostatic mechanisms. These findings validate S333A and S333E as mechanistically distinct but functionally informative proxies for studying phosphorylation's effects, with the structural gradient (WT<S333A<S333E) paralleling the functional disruption observed in our GTPase assays and cell length measurements. The structural similarities between S333E and phospho-S333 strongly suggest that phosphorylation would disrupt core binding at least as effectively as S333E, with the phosphate's greater negative charge and steric bulk likely enhancing this disruption.

## Tail-core binding modulates access to FtsZ regulators

The FtsZ core's C-terminal polymerization surface is targeted by negative regulators of division. Having shown that the tail binds to this same surface, we wondered whether tail-core binding might regulate inhibitor access. To test this hypothesis, we focused on MinC, the classic regulator of division site placement. We noticed that the tail's binding site on the core (H10/S9 and T7/H8) overlaps with the previously characterized binding site of MinC$_N$, the 92aa N-terminal domain of MinC [35] (Fig 4A-B, S12). We therefore used this protein fragment as a spatial probe in competitive binding experiments.

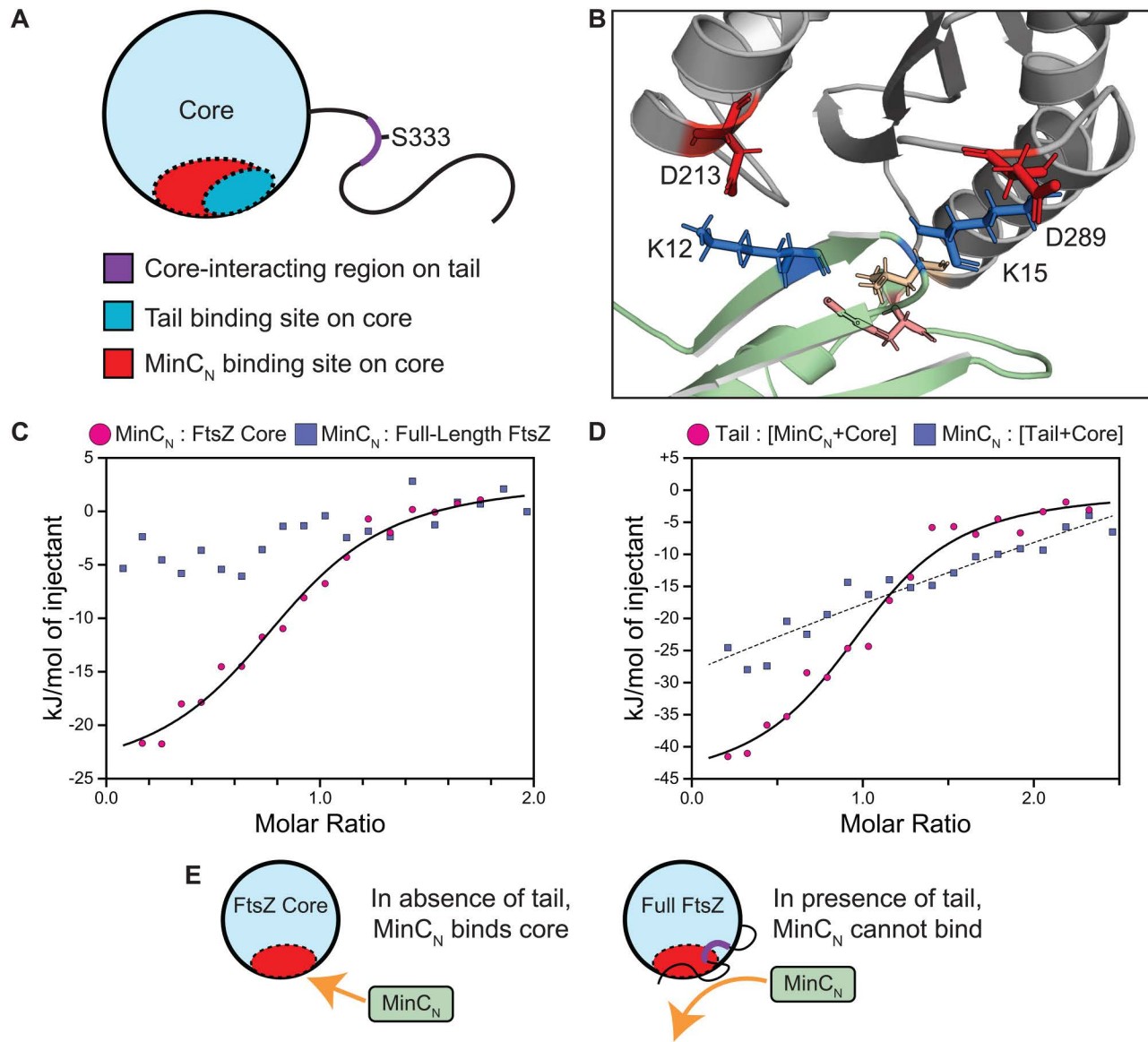

**Fig. 4. The FtsZ tail binds the C-terminal polymerization surface and gates MinC$_N$ access. (A)** Schematic representation of the FtsZ core highlighting the known MinC$_N$ binding site (red) and the predicted FtsZ tail binding site (blue), which partially overlap. The tail is depicted with its core-binding region (L330-H337, purple) and the S333 phosphosite. **(B)** AlphaFold model of the MinC$_N$-FtsZ core interface, adapted from Machado et al. 2025 [35]. MinC$_N$ is shown in pale green. Ionic interactions occur between MinC$_N$ residues K12 and K15 (marine) and FtsZ residues D213 and D289 (red), respectively. MinC$_N$ helix H1 (wheat) makes an additional hydrophobic contact with FtsZ helix H10 (salmon). **(C)** ITC demonstrates that the tail occludes MinC$_N$ binding. MinC$_N$ binds specifically to the isolated FtsZ core (magenta, $K_d = 495 \pm 110$ nM, N = 1 ± 0.15) but shows no detectable binding to full-length FtsZ (blue, $K_d > 2$ μM), indicating the tethered tail blocks MinC$_N$ access. **(D)** Competitive displacement experiments via ITC. The FtsZ tail can displace prebound MinC$_N$ from the core (magenta), whereas MinC$_N$ cannot displace pre-bound tail from the core (blue), demonstrating that the tail dominates access to the polymerization surface. **(E)** Model summarizing tail-mediated regulation of MinC$_N$ access. The MinC$_N$ binding site on the FtsZ core is denoted in red; the core-binding region of the tail is highlighted in purple. In its unphosphorylated state, the tail occupies the core's polymerization surface, blocking MinC$_N$ (pale green) binding. S333 phosphorylation releases the tail, exposing the MinC$_N$ binding site.

We performed ITC experiments, titrating $MinC_N$ into either isolated FtsZ core (lacking its tail) or full-length FtsZ (including its tail). As expected, $MinC_N$ exhibits specific binding to the tailless core ($K_d = 495 \pm 110$ nM, N = 1 ± 0.15; Fig 4C), consistent with previous studies [35]. However, $MinC_N$ binding was abolished when titrated into full-length FtsZ ($K_d > 2$ µM; Fig 4C). Since the tail is normally tethered to the core, this surprising result suggests two possibilities: either tail-core binding specifically occludes the $MinC_N$ binding site, or the mere presence of the tail blocks $MinC_N$ access through some other mechanism. A direct occlusion model is consistent with the tail's two competitive advantages: two-fold higher affinity for the core ($K_d = 228$ nM vs 405 nM for $MinC_N$) and an enormous effective concentration advantage when tethered intramolecularly.

To distinguish between these possibilities, we tested whether our tail mutations that disrupt core binding would restore $MinC_N$ access. $MinC_N$ bound to full-length FtsZ variants containing S333A and S333E tail substitutions with affinities comparable to the tailless core (S13A-B Fig). On the other hand, the full-length K335A/H337A double mutant showed binding of multiple $MinC_N$ molecules per FtsZ (S13C Fig), indicating significantly altered engagement of $MinC_N$ with FtsZ. These results demonstrate that the wild-type tail directly occludes $MinC_N$'s binding site on the polymerization surface and can alter how molecules such as $MinC_N$ engage FtsZ. Furthermore, they establish that each tail binds its own core intramolecularly (*in cis*), validating the physiological relevance of the *trans* binding we characterized in Fig 2A. This intramolecular tail-core interaction prevents binding of exogenous tails, as demonstrated by the inability of isolated wild-type tail to bind full-length FtsZ ($K_d > 2$ µM; S13D Fig).

To further probe the competitive relationship between the tail and $MinC_N$, we performed displacement experiments. First, we asked whether the tail could displace $MinC_N$ from the core. We pre-formed a $MinC_N$:Core complex by mixing 45 µM $MinC_N$ with 5 µM Core (9:1 ratio), achieving ~99% occupancy based on our measured $K_d$. Upon titrating Tail to a final concentration of 10 µM, we observed robust binding with 1:1 stoichiometry (Fig 4D). This demonstrates that the tail can completely displace $MinC_N$ from the core, despite starting from a saturated complex.

We then tested the reciprocal experiment: can $MinC_N$ displace a pre-bound tail? We pre-formed a Tail:Core complex using 30 µM Tail and 5 µM Core (6:1 ratio), again achieving ~99% occupancy. When we titrated $MinC_N$ to 10 µM, we observed no detectable binding (Fig 4D). To ensure a fair comparison, we had adjusted the ligand:core ratios to account for their different affinities—using 9:1 for $MinC_N$:Core versus 6:1 for Tail:Core—so that both complexes started at equivalent occupancy. Under these conditions, the tail completely displaced $MinC_N$, while $MinC_N$ could not detectably displace the tail. Together, these competition experiments demonstrate that the tail dominates access to the polymerization surface, effectively gating $MinC_N$'s access to the core (Fig 4E).

## Discussion

While FtsZ's C-terminal tail has primarily been viewed as an inert mechanical tether, our data demonstrate that it also serves as a regulatory element through direct intramolecular interaction with FtsZ's globular core (Fig 5). We identified a molecular recognition element (L330-H337) centered on S333 that mediates high-affinity binding to the core's C-terminal polymerization surface. Phosphorylation of S333 by PrkC acts as a molecular switch: it drives the I334 and K335 side-chains together, sequestering them from their binding sites on the core and thereby exposing the polymerization surface. This phospho-switch mechanism explains both the enhanced intrinsic polymerization we observed in S333 mutants and their ability to restore $MinC_N$ binding. The context-dependent cell length phenotypes—shorter in minimal media, longer in rich media—suggest this regulatory mechanism may coordinate FtsZ's response to different growth conditions through differential effects on division regulators.

This autoinhibitory mechanism operates through direct competition with other FtsZ monomers for the C-terminal polymerization surface. This surface is exposed at the growing end of FtsZ filaments where new monomers must bind to extend the polymer. Tail binding transiently occludes this addition site, explaining why S333 mutations that disrupt tail-core binding reduce the critical concentration for polymerization. However, this regulation occurs through dynamic equilibrium rather

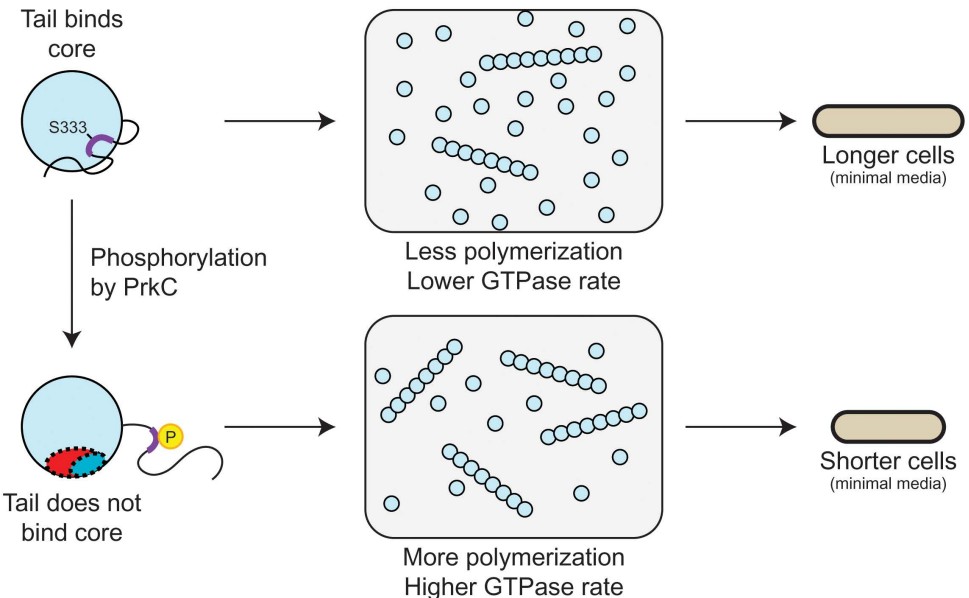

**Fig 5. Tail-core binding regulates FtsZ activity.** Diagram illustrating how S333 phosphorylation affects FtsZ dynamics and cell physiology across multiple scales. **Top row (unphosphorylated state):** The FtsZ tail's L330-H337 region binds the core's C-terminal polymerization surface, competing with monomer addition and occluding regulatory proteins. This results in higher critical concentration, reduced GTPase activity, and longer cells in minimal media. **Bottom row (phosphorylated state):** PrkC phosphorylation of S333 induces I334-K335 compaction, disrupting tail-core binding. This exposes the polymerization surface, enabling enhanced monomer addition and regulatory protein access, leading to lower critical concentration, increased GTPase activity, and shorter cells in minimal media. The model demonstrates how a single phosphorylation event modulates both intrinsic polymerization dynamics and extrinsic regulatory control.

than a static blockade: dissociation of the tail creates windows for monomer addition. Once a monomer incorporates, its binding is essentially irreversible on the timescale of tail-core interactions; the penultimate subunit's tail can no longer rebind its now-occupied C-terminal surface. S333 mutations shift this equilibrium toward the unbound state, increasing monomer access to filament ends; our structural analysis shows that phosphorylation would do the same. The resulting increase in polymer abundance directly explains the enhanced GTPase activity we observed, since hydrolysis occurs at monomer-monomer interfaces within filaments.

Furthermore, our data indicate that the FtsZ tail gates access to the C-terminal polymerization surface in a regulator-specific manner. Our competitive binding experiments demonstrate that the tail blocks MinC$_N$ binding *in cis* and can even displace pre-bound MinC$_N$ *in trans*—yet MinC$_N$ cannot reciprocally displace the tail. This asymmetric competition, combined with our mutational analysis showing that tail and MinC$_N$ share overlapping binding sites, establishes the tail as a regulator of polymerization surface accessibility. S333 modifications that disrupt tail-core binding restore full MinC$_N$ access, confirming that the tail directly gates MinC$_N$ binding. However, this gating appears to be selective rather than universal. For example, MciZ has been reported to bind to full-length FtsZ [48], indicating the tail does not occlude its binding site despite targeting the same general surface. This differential occlusion suggests the tail acts as a selective gate—blocking certain regulators like MinC$_N$ while permitting others like MciZ. S333 phosphorylation releases this gate, simultaneously exposing the polymerization surface to FtsZ monomers (enhancing assembly) and sensitive regulators like MinC$_N$ (enabling inhibition).

The tail's ability to occlude MinC$_N$ suggests it may also gate other regulators, which could explain the nutrient-dependent cell length phenotypes we observed. In minimal media, S333 mutations produce 5–6% shorter cells, while

the same mutations in rich media produce 5–7% longer cells. This reversal likely reflects the differential activity of UgtP, a metabolic regulator of FtsZ that responds to nutrient availability [43]. In minimal media, where UgtP forms inactive aggregates in the cytoplasm [49], the enhanced FtsZ polymerization from our S333 mutations produces shorter cells—consistent with previous work showing that doubling FtsZ expression reduces cell length by 10% [42]. In rich media, UgtP localizes to the division site where it inhibits FtsZ assembly by capping or severing filaments [49]— suggesting this regulator may also bind the core's C-terminal polymerization surface. S333 mutations that expose this surface would enhance UgtP's inhibitory access, potentially explaining longer cells despite FtsZ's increased polymerization potential. This model positions S333 as an integration point for both spatial (MinC) and metabolic (UgtP) regulatory inputs.

Our findings reveal S333 phosphorylation as a regulatory switch coordinating multiple aspects of Z-ring assembly through a single modification. By controlling tail-core binding, phosphorylation simultaneously modulates FtsZ polymer abundance through: (1) intrinsic effects on monomer competition, and (2) extrinsic effects via regulatory protein access. This mechanism integrates diverse regulatory inputs at a single phosphosite. In its unphosphorylated state, the tail dominates the core's C-terminal polymerization surface; phosphorylation reverses this dominance, enhancing intrinsic polymerization while exposing the surface to division inhibitors. This dual effect—enhanced assembly potential coupled with increased inhibitor sensitivity—creates a highly regulable state where division can proceed rapidly when permitted but is readily blocked when conditions deteriorate. S333 phosphorylation may thus serve as a molecular rheostat, fine-tuning the balance between intrinsic polymerization potential and susceptibility to inhibitory regulators.

This mechanism builds on previous observations of PrkC-mediated FtsZ phosphorylation. While phosphoproteomics studies have detected this modification in *B. subtilis* [37–39], the precise site and functional consequences remained unclear. We demonstrate that PrkC phosphorylates FtsZ specifically at S333, where the tail binds the core's polymerization surface, and we reveal the structural mechanism by which phosphorylation disrupts this interaction: it drives I334 and K335 together, sequestering these key binding residues from their core partners. While FtsZ phosphorylation by PrkC homologs has been documented in other Gram-positive species (Table A in S1 Appendix), previous studies focused on measuring bulk polymerization changes, overlooking the tail-core regulatory mechanism and the structural basis for phosphorylation-induced disruption. Our atomic-level structural analysis now provides a testable framework for investigating whether similar mechanisms operate across diverse Gram-positive species employing PrkC-FtsZ signaling.

The involvement of PrkC embeds the S333 phospho-switch within a broader cellular signaling network. PrkC is known to link growth and division processes to environmental conditions across diverse bacteria [33]. Its activity is regulated by extracellular PASTA domains [50], which bind muropeptides released during normal peptidoglycan turnover or stress conditions like β-lactam antibiotic exposure [50,51]. Upon activation, PrkC coordinates growth through phosphorylation of targets involved in peptidoglycan synthesis (RodZ [52]), cell wall remodeling (WalRK [53]), and protein synthesis (EF-Tu [54], EF-G [55]). Our work now establishes FtsZ as a functional member of this network, explaining phosphorylation events detected in prior phosphoproteomic studies. PrkC homologs are found primarily in Gram-positive bacteria [51,56], where their PASTA domains are directly exposed to the environment due to the absence of an outer membrane. This distribution suggests these kinases evolved as specialized stress sensors for organisms where the PG cell wall serves as the primary interface with the environment.

Beyond bacteria, our structural analysis provides atomic-level insight into how phosphorylation might control IDR interactions more broadly, addressing a significant gap in our understanding of kinase signaling. While kinases typically target disordered regions—loops, linkers, and tails—the structural consequences of IDR phosphorylation remain poorly understood due to the dynamic nature of these regions [57]. Our work elucidates one such strategy: phosphorylation can structurally disrupt molecular recognition elements within IDRs to control autoinhibitory interactions. Our findings

also challenge the conventional interpretation of phospho-dead/-mimetic mutants. This binary thinking—where S333A preserves 'the unphosphorylated state' and S333E mimics 'the phosphorylated state'—would predict opposite functional outcomes. Instead, our structural analysis reveals why these FtsZ mutations produce a gradient of effects: they disrupt the same tail-core interaction to varying degrees, through different mechanisms. The functionally relevant parameter is not phosphorylation state *per se*, but whether the tail can bind the core. Crucially, this mechanistic insight required structural analysis; functional assays alone would have treated both as simple loss-of-function mutants. As phosphorylation of disordered regions is increasingly recognized as a central regulatory mechanism across all domains of life, characterizing the structural impact of these modifications will be essential for understanding how they work.

## Methods

### Culture conditions

*Bacillus subtilis* strains were stored as glycerol stocks at −80°C. For routine growth, cells were streaked onto Luria-Bertani (LB) agar plates containing appropriate antibiotics and incubated overnight at 37°C. For experimental cultures, cells were grown in the chemically defined S7 minimal medium [58] with the following modifications: 50 mM MOPS buffer (instead of the original 100 mM) [59], 20 mM potassium glutamate (instead of sodium glutamate), and 1% (w/v) glucose.

### Strain construction

Genetic constructs were generated using a multi-step assembly strategy involving polymerase chain reaction (PCR) [60], Overlap Extension PCR [61], and Gibson assembly [62,63]. Constructs were designed in Geneious [64] and typically involved anywhere from two to six fragments. Individual DNA fragments were amplified via standard PCR, then stitched together using Overlap Extension PCR to create 2–3 intermediate fragments of approximately equal size. These were assembled into complete genetic constructs via Gibson assembly, with approximately 1 kb homology arms at each end.

 *B. subtilis* strains were derived from the prototroph PY79 [65]. To prepare competent cells, single colonies were inoculated into 1 mL of MC medium [66] and grown at 37°C with rolling aeration to an optical density ($OD_{600}$) of 1.10–1.30 (approximately 4 hours). Competent cells (200 μL) were transformed with either 20 μL of Gibson assembly reaction or 1 μL of purified genomic DNA and incubated with rolling at 37°C for 2 hours. Transformant cultures (50 μL) were then plated on appropriate selective media and incubated overnight at 37°C. Isolated colonies were streak-purified on selective plates with overnight incubation at 37°C. For long-term storage, single colonies were grown in LB medium to late logarithmic phase, and 20% glycerol stocks were prepared and stored at −80°C.

 Strains generated for this study are listed in Tables C and D in S1 Appendix; oligonucleotides are listed in Table F in S1 Appendix. All strains were validated via Sanger (Genewiz) or Nanopore (Plasmidsaurus) sequencing.

### Plasmid construction

Expression plasmids for protein purification were constructed as follows: Full-length *B. subtilis* FtsZ and variants were cloned into pET-28a with N-terminal 6xHis tags and TEV cleavage sites. FtsZ tail constructs (residues 316−382) were cloned into pGEX-6P-3 with N-terminal GST tags and PreScission protease cleavage sites. $MinC_N$ was PCR amplified from *B. subtilis* genomic DNA and cloned into pGEX-6P-1 with an N-terminal GST tag and a PreScission protease cleavage site. Point mutations were introduced using the Q5 Site-Directed Mutagenesis kit (NEB) with primers designed using the NEBaseChanger tool. Detailed construction methods, including restriction sites, cloning strategies, and insert sequences, are provided in S1 Appendix. Plasmids generated for this study are listed in Table E in S1 Appendix; oligonucleotides are listed in Table F in S1 Appendix. All plasmids were validated via Nanopore sequencing (Plasmidsaurus).

## Protein purification

One Shot® BL21 Star™ (DE3) Chemically Competent *E. coli* (Thermo) cells containing $MinC_N$ or FtsZ constructs were inoculated in LB media containing 100 µg/mL kanamycin and grown overnight at 37°C with shaking at 200 rpm. In the morning, 20 mL of overnight culture was diluted into 2 L LB media containing 100 µg/mL kanamycin. Cells were allowed to grow at 37°C with shaking until reaching $OD_{600} \sim 1.2$ where they were induced with 100 µM IPTG for 2 hours, after which they were pelleted at 4,000 × g for 20 minutes. A pellet corresponding to 2 L was resuspended in TKEG buffer (50 mM Tris pH 8, 50 mM KCl, 1 mM EDTA, and 5% glycerol) and sonicated using a 3/8" tip (200 Watt Ultrasonic Homogenizer) at a pulse-time setting with 60% amplitude for 8 rounds of 15 seconds on and 30 seconds off for each sample. Lysate was cleared by spinning at 21,000 × g for 20 minutes.

Lysate supernatant was added to Clontech Labs 3P TALON® Metal Affinity Resin and allowed to bind for 2 hours on a rocker at 4°C. Following binding, beads were washed twice with 100 mL of TKEG buffer on a column. Protein-bead complexes were placed in a 50 mL Falcon tube in 10 mL of HMK buffer (50 mM HEPES, 100 mM KAc, 5 mM MgAc, pH 7.7) and 15 µL of Novex AcTEV Protease. The column was allowed to rock at 4°C for at least 8 hours prior to collecting the eluate.

For $^{15}N$-labeled FtsZ tail purification used in NMR studies, overnight cultures were inoculated and grown in LB up to an OD of 0.4 and pelleted at 4,000 × g for 20 minutes. Cell pellets were resuspended in pre-warmed M9 minimal medium and growth was continued as noted above.

## Western blot analysis

Strains expressing FtsZ variants (Table C in S1 Appendix) were grown in 1 mL S7 minimal medium (modified with 50 mM MOPS and 50 µM iron citrate) at 37°C to mid-logarithmic phase ($OD_{600} \sim 0.5$), normalized to $OD_{600}$ 0.05, grown again to $OD_{600} \sim 0.5$, normalized again to $OD_{600}$ 0.05 in 5 mL S7 medium, and grown to $OD_{600}$ 0.2. Pellets were harvested by centrifugation (5,000 × g, 10 minutes, 4°C), washed once with lysis buffer (20 mM Tris-HCl pH 7.5, 10 mM EDTA pH 8.0, 100 mM NaCl, 1 mM PMSF), and stored at −80°C overnight.

Cell pellets were resuspended in 50 µL lysis buffer supplemented with 1 mg/mL lysozyme, 10 µg/mL DNase I, and 100 µg/mL RNase A and incubated for 20 minutes at 37°C. Total protein concentration was determined using a NanoDrop 1000 spectrophotometer (Thermo) via A280. Lysates were mixed with Laemmli buffer containing 10% β-mercaptoethanol and 1 mM DTT, heated at 95°C for 10 minutes, and centrifuged (15,000 × g, 5 minutes, RT).

Protein samples (50 µg per lane) were loaded on a 10% SDS-polyacrylamide gel and separated at 120V for 1.5 hours. Proteins were transferred to a 0.2 µm nitrocellulose membrane (BioRad) using a Trans-Blot Turbo Transfer system (BioRad) at 25 V (1.0 A limit) for 30 minutes in Trans-Blot Turbo Transfer Buffer (BioRad). Membranes were stained with Ponceau S for 15 minutes at RT with shaking, rinsed five times with water until white, and imaged with a BioRad GelDoc. Membranes were then blocked with TBST (TBS with 0.1% Tween-20) with 5% non-fat milk for 30 minutes at RT, then incubated with rabbit polyclonal anti-FtsZ antibody (1:1,000 dilution; a gift from Jeff Errington) overnight at 4°C. After washing three times (10 minutes each) with TBST, membranes were incubated with HRP-conjugated goat anti-rabbit secondary antibody (1:10,000 dilution; LI-COR IRDye 680LT Goat anti-Rabbit IgG) for 2 hours at RT. After three TBST washes (10 minutes each), immunoreactive bands were imaged with a BioRad GelDoc. Band intensities were quantified using Python.

## PhosTag gels

Strains expressing HaloTag-FtsZ variants (Table C in S1 Appendix) were grown in S7 minimal media (modified with 50 mM MOPS) with 20 µM IPTG at 37°C to $OD_{600}$ 0.5. Cultures were labeled with 50 nM HaloTag TMR (tetramethylrhodamine) ligand (Promega) for 30 minutes at 37°C (final $OD_{600}$ 0.7), harvested by centrifugation (5,000 × g, 5 minutes, 4°C), and stored at −80°C.

Cell pellets were lysed in 50 mM Tris-HCl pH 7.5, 150 mM NaCl, 3 mM MgCl2, supplemented with protease and phosphatase inhibitors (1 mM PMSF, 50 mM NaF, 1 mM Na3VO4), 200 µg/mL lysozyme, and 1 µL SAN nuclease (ArcticZyme). After 30 minutes at 37°C, lysates were mixed with Laemmli buffer containing 10% β-mercaptoethanol and 1 mM DTT, heated at 95°C for 10 minutes, and centrifuged (15,000 × g, 5 minutes, RT).

Phosphorylation was analyzed using SuperSep Phos-tag gels (12.5%, Wako Chemicals). For each sample, eight 2-fold serial dilutions were prepared in sample buffer. Samples (10 µL) were separated alongside a molecular weight marker by electrophoresis in Tris-glycine-SDS running buffer at 15 mA (150V limit) for 3.5 hours. HaloTag-TMR labeled proteins were visualized using an Azure Sapphire biomolecular imager with the 520 nm laser line. Per-condition replicate counts were as follows: WT, n = 3; S333A, n = 2; ΔPrkC, n = 2; S329A-S333A, n = 1.

## Microscopy: Culture conditions

Single colonies from LB overnight plates were inoculated into modified S7 medium supplemented with 50 µM iron citrate [67,68], and 4-fold serial dilutions were incubated overnight at 37°C with rolling aeration. This ensured all cultures underwent a minimum of 13 generations of growth before imaging, allowing cells to reach steady-state size after transitioning from stationary phase on LB plates to logarithmic growth in minimal medium [69]. The following morning, cultures that had reached an $OD_{600}$ between 0.10–0.30 were selected and further serial diluted into fresh S7 medium, and ten 2-fold serial dilutions were prepared for the day's imaging. These cultures were grown at 37°C with rolling aeration for at least 4 hours prior to experimentation. All imaging was performed in early log phase, at $OD_{600}$ 0.12–0.25.

## Microscopy: Sample preparation

Cultures were removed from 37°C and stained with FM 5-95 dye (1 µg/mL final, Thermo) by gentle swirling followed by 1 minute incubation at room temperature. Cells were immobilized by spotting 1 µL of stained culture onto a NaOH-washed coverslip and overlaying with a pad of S7 medium containing 2% agarose and FM 5-95 dye (0.5 µg/mL). Agarose pads were prepared by melting agarose in S7 medium, adding FM 5-95 dye, and casting between glass plates using plastic frames (1.5 cm × 1.5 cm × 1 mm) as spacers.

## Microscopy: Flat field correction

Flat field correction was performed using 100 µM sodium fluorescein. For each correction set, 100 fields were collected without illumination (dark fields) and with 450 nm illumination through a FITC filter cube (flat fields). Dark and flat field composites were generated by averaging the respective 100-field sets. A pixel-wise gain map was then calculated as

$$\mathrm{mean}(Flat - Dark) \, / \, (Flat - Dark)$$

Images were corrected by dark subtraction followed by multiplication with the gain map.

## Microscopy: Epifluorescence imaging

Epifluorescence images were collected on a Nikon Ti-E microscope using a Nikon Plan Apo 100X Oil Ph3 objective (1.40 NA) with an Andor Clara DR-2136 CCD camera. FM 5-95 was excited at 550 nm using a CoolLED pE-300 Ultra LED source and imaged through a TRITC filter cube modified with an ET610lp emission filter (Chroma) to account for FM 5-95's unusually large Stokes shift.

## Cell length measurement

Images were segmented using Omnipose v0.5 [70], followed by manual correction of segmentation errors in Napari v0.4 [71]. Cell lengths were measured using custom Python code: Adjacent cell masks were merged into chain masks. Chain

and cell boundaries were extracted from their masks using the marching squares algorithm, then refined with an active contour model aligning with membrane signals. Chain midlines were constructed by finding the longest path through Voronoi ridge vertices interior to the chain boundary. Chain midlines were then bisected with cell boundary contours to obtain cell midlines. Cell lengths were determined by measuring along their midline skeletons.

Analysis was restricted to cells in chains of three or more, as single and chained cells exhibited systematically different length distributions. To control for batch effects and biological variability, each strain was imaged multiple times, both within and between days. To account for the hierarchical nature of this experimental design, each replicate (approximately 100–300 cells) was collapsed to its median cell length. Each strain was represented by the arithmetic mean of these replicate medians, and statistical significance was assessed using a two-tailed Student's t-test performed on the replicate medians.

### AlphaFold analysis

The FtsZ core (residues 1–315) and CTL (residues 316–365) sequences were co-folded using AlphaFold2-Multimer v2.3 followed by Amber relaxation [45,72]. 100 models were generated (20 random seeds × 5 models per seed). Predicted aligned error (PAE) matrices were analyzed to assess prediction confidence at the core-CTL interface. PAE is shown as both a pairwise distance matrix and as a line plot of minimum PAE values per residue (S6 Fig). Interface residues were identified based on proximity (≤3 Å) between core and CTL atoms. Structural analysis and visualization were performed using PyMOL [73].

### Analysis of geometric constraints on intramolecular binding

To measure the distance from the tail exit point to the center of the MIF pocket, we used PyMOL's distance command to measure the Euclidean Cα-Cα distance between G314 and I293 on an X-ray crystal structure of the FtsZ core (PDB: 2VAM [47]). To estimate the reach of the 20aa linker (G314-S333), we applied the polymer scaling relationship $R = R_0 \times N^\nu$, where $R_0 = 3.8$ Å (Cα-Cα distance per residue), $N = 20$ residues, and $\nu = 0.5$ (ideal chain) or $0.6$ (self-avoiding chain), yielding a characteristic end-to-end distance of 17–23 Å.

### Measurement of GTPase activity

FtsZ GTPase activity was measured using a malachite green phosphate assay. Proteins were purified via His6 affinity purification and eluted in HMK buffer (50 mM HEPES, 5 mM magnesium acetate, 100 mM potassium acetate, pH 7.7). Cellular phosphate was removed via buffer exchange into HMK buffer using Sephadex G-25 columns (Cytiva). Protein concentration was determined using a Pierce BCA Protein Assay Kit (Thermo). FtsZ was diluted to an initial concentration of 4 µM, followed by linear serial dilutions in one-eighth steps in HMK buffer.

FtsZ dilutions were kept on ice. GTPase reactions were initiated by the addition of GTP to a final concentration of 1 mM in a total reaction volume of 50 µL. Reactions were incubated at 37°C for 15 minutes in a PCR thermocycler. Following incubation, reactions were immediately quenched by 20-fold dilution into ice-cold 10 mM EDTA (10 µL of reaction into 190 µL of EDTA). The diluted samples were split into duplicate 100 µL aliquots. Using a BioMek FXp liquid handling system (Beckman Coulter), 80 µL of each diluted sample was simultaneously transferred into 20 µL of malachite green reagent (Malachite Green Phosphate Assay Kit, MAK307, Sigma) in a 96-well clear-bottom plate. Plates were incubated for 30 minutes at RT, during which the plates were spun at 3,000 × g for 15 minutes to eliminate bubbles. Absorbance was measured at 620 nm using an Epoch2 microplate spectrophotometer (BioTek). Phosphate concentrations were determined using phosphate standards (0–40 µM) prepared in 10 mM EDTA to match the buffer used for reaction quenching and sample dilution.

The phosphate production rate was calculated by dividing the total measured phosphate by the reaction time (15 minutes) and multiplying by the dilution factor (20). Since hydrolysis of one GTP molecule produces one inorganic phosphate, this phosphate production rate directly reflects the rate of GTP hydrolysis. For each FtsZ variant, GTPase rate was plotted

against protein concentration. The slope of this line represents the specific activity (GTP/min/FtsZ), and the x-intercept defines the critical concentration (µM). Each variant was analyzed in a minimum of 3 independent experiments run on different plates.

## Statistical analysis of GTPase activity and critical concentration

Data analysis was performed in Python. For both GTPase activity and critical concentration, bootstrap analysis was performed by randomly selecting one measurement at each concentration and performing linear regression. This process was repeated 1,000 times to generate distributions of slopes and x-intercepts. Means and 95% confidence intervals were calculated from these distributions. Statistical significance was assessed by comparing bootstrap distributions between conditions using two-tailed Student's t-tests, with differences considered significant at $p < 0.001$.

## Isothermal titration calorimetry

ITC experiments were performed in TKEG buffer (50 mM Tris pH 8, 50 mM KCl, 1 mM EDTA, and 5% glycerol) at 25°C. Full-length FtsZ or FtsZ core were diluted to 5 µM; FtsZ tail and $MinC_N$ were diluted to 50 µM. For competition experiments, protein complexes were pre-formed by mixing ligand and core at ratios calculated to achieve ~99% occupancy based on measured $K_d$ values, then immediately loaded into the ITC cell. For $MinC_N$ displacement: 45 µM MinCN + 5 µM Core (9:1). For Tail displacement: 30 µM Tail + 5 µM Core (6:1).

ITC experiments were performed on a TA Instruments Affinity ITC by loading full-length FtsZ, FtsZ core, or pre-formed complexes into the cell, and FtsZ tail or $MinC_N$ into the syringe. The cell temperature was equilibrated to 25°C with a stirring speed of 125 rpm and an initial delay of 600 seconds. Titrants were injected in 2 µL increments for a total of 20 injections with 180 second intervals between injections. Baseline correction, binding curve fitting, and parameter calculation were performed using NanoAnalyze. All ITC experiments were performed in at least duplicate. Binding parameters for all replicates are provided in S1 Table.

## Nuclear magnetic resonance spectroscopy

NMR spectra were acquired on a Bruker Avance III HD 800 MHz spectrometer equipped with a TCI cryoprobe at 298K. Samples were prepared in 50 mM HEPES-$d_{18}$ (98% deuterated) pH 7.5, 50 mM KCl, 1 mM EDTA in 90% $H_2O$/10% $D_2O$. Chemical shifts were referenced to DSS. Spectra were processed with NMRPipe [74] and analyzed with Poky [75].

## NMR: Data processing

Time-domain data were zero-filled and apodized with squared sine bell functions prior to Fourier transformation. For consistent quantitative comparison across samples, phase corrections were systematically optimized. NOESY spectra were phased to minimize row-wise intensity variance in the fingerprint region (7.85–8.50 ppm in F2; 0.5–5.0 ppm in F1), reducing artifacts from imperfect phase correction. HSQC spectra were phased to maximize the ratio of positive to negative signal intensity in the backbone amide region (7.85–8.50 ppm in F2; 113–127 ppm in F1).

## NMR: Peptide variant experiments

Synthetic 15aa peptides corresponding to the FtsZ tail's core-binding region (P328-P342) were synthesized and HPLC-purified to >95% purity at the Tufts University Core Facility (TUCF). Four variants were prepared: wild-type, S333A, S333E, and phospho-S333. Natural abundance $^1H$-$^{15}N$ HSQC and $^1H$-$^1H$ NOESY spectra were collected. NOESY experiments used a mixing time of 120 ms with excitation sculpting for water suppression. Backbone amide resonances were assigned using sequential NOE walks and amino acid-specific patterns in NOESY spectra, using characteristic chemical shift values from the BMRB database [76] to predict approximate peak locations for each residue type. These

assignments were then transferred to HSQC spectra via shared $^1$H chemical shifts. HSQC peak assignments are provided in S4 Table. NOESY peak assignments are provided in S6 Table.

### NMR: Chemical shift perturbation analysis

Chemical shift perturbations (CSPs) were calculated to quantify backbone structural changes upon S333 modification. To correct for sample-to-sample variation, spectra were aligned using three reference residues distant from the modification site (S329, Q339, S340). For each variant, reference residue $\Delta\delta_H$ were calculated relative to wild-type, averaged to obtain a global offset, and subtracted from all peaks. After alignment, CSPs were calculated for each variant peptide (S333A, S333E, and phospho-S333) relative to wild-type using the standard formula:

$$CSP = \sqrt{[(\Delta\delta_H)^2 + (\Delta\delta_N/5)^2]}$$

where $\Delta\delta_H$ and $\Delta\delta_N$ represent the chemical shift differences in the $^1$H and $^{15}$N dimensions, respectively. The scaling factor of 5 accounts for the difference in chemical shift dispersion between $^1$H and $^{15}$N nuclei. All CSP values are provided in S5 Table.

### NMR: Peak intensity normalization

To account for differences in peptide sample concentration and acquisition time, peak heights were normalized using the median of selected peaks. For HSQC spectra, all backbone amide peaks excluding positions 333 (modification site) and 337/338 (overlapped peaks) were used. For NOESY spectra, diagonal peaks were excluded due to their high intensity and susceptibility to overlap artifacts. All off-diagonal intraresidue (i,i) and sequential (i,i+1) NOEs observed consistently in all four samples were used. For NOEs annotated on both sides of the diagonal, peak heights were averaged to reduce measurement noise. Peak heights were divided by these median values and divided by $10^6$ for convenient numerical display.

For spectral display, contour levels were adjusted to maintain comparable signal across samples. A baseline contour threshold was set just above the noise floor in the wild-type spectrum (as determined by visual inspection), then divided by each sample's normalization factor to establish equivalent contour thresholds for variant peptides. Contour level multipliers were kept constant at 1.4 across all spectra. Normalization factors and resulting contour thresholds are provided in S7 Table.

### NMR: High-resolution NOESY cross-sections

To visualize changes in K335 Hβ2/Hβ3 cross-peak patterns, NOESY spectra were reprocessed with zero-filling to 4,096 points in the indirect dimension (8×increase) to enhance digital resolution. One-dimensional cross-sections were extracted at the K335 HN chemical shift using nmrglue [77] and normalized as described above.

### NMR: Tail-core binding experiments

$^{15}$N-labeled FtsZ tail (67aa) and unlabeled FtsZ core (315aa) were purified. $^1$H-$^{15}$N HSQC spectra were collected using a TROSY-based pulse sequence with WATERGATE water suppression. Spectra were acquired for $^{15}$N-labeled tail alone and in complex with unlabeled core at a tail:core molar ratio of 1:1.

### NMR: Tail-core binding quantification

To quantify changes in the 67aa tail upon core binding, peaks were manually picked using Poky [75]. Tail-core peaks were matched to the nearest free tail peak by CSP distance (threshold: 0.10 ppm; ties resolved by closest match). Spectra were

normalized using median heights of stable reference peaks (defined by |log$_2$ fold-change| < 0.5), with all intensities scaled relative to free tail. Chemical shift perturbations and intensity ratios were calculated for each paired peak. Residues were assigned by matching tail peaks to peptide peaks by CSP distance (threshold: 0.05 ppm; ties resolved by closest match), yielding eight assigned residues: L330, Q332, S333, I334, K335, H337/N338, and V341. All peak assignments are provided in S2 Table; binding analysis is provided in S3 Table.

## Software and data analysis

All data processing, computational image analysis, and statistical analysis was performed using Python 3 [78] with the NumPy [79], SciPy [80], Pandas [81], and Scikit-Image [82] libraries. Plots were generated using Matplotlib [83] and Seaborn [84]. Gels were analyzed using Fiji [85]. NMR spectra were processed with NMRPipe [74], analyzed with Poky [75], and visualized with nmrglue [77]. Protein structural models were generated with AlphaFold [72] and visualized with PyMOL [73].

## Supporting information

**S1 Raw Image. 2025.10.26 S1 raw images.**
(PDF)

**S1 Fig. FtsZ core structural features.** Crystal structure of the *B. subtilis* FtsZ core (PDB: 2VAM [47]). The core comprises an N-terminal domain (light blue), a C-terminal domain (pale green), and an interdomain helix H7 (magenta). Key secondary structural elements (turn T7, α-helices H8 and H10, β-strand S9) are labeled. The polymerization axis runs vertically, with subunits assembling head-to-tail via their N-terminal (top) and C-terminal (bottom) polymerization surfaces. GTP binds at the N-terminal surface of one monomer; upon polymerization, the adjacent monomer's C-terminal surface contributes D213 to complete the active site, positioning this catalytic residue for eventual GTP hydrolysis within filaments. The tail exits the core from the side of the C-terminal domain.
(EPS)

**S2 Fig. PhosTag gel analysis of FtsZ phosphorylation.** PhosTag gel analysis of HaloTag-FtsZ proteins labeled with TMR, visualized using an Azure Sapphire biomolecular imager with 520 nm excitation. Samples were prepared from cells grown to mid-log phase (OD$_{600}$ 0.7) in S7 minimal medium. (A) Four 2-fold serial dilutions each of wild-type, S333A, and ΔPrkC strains. The second dilution from each strain is shown in Fig. 1B. (B) Eight 2-fold serial dilutions of wild-type and S329A-S333A double mutant strains, demonstrating that mutation of both putative phosphosites eliminates the upper phosphorylated band. (C) Eight 2-fold serial dilutions of S329A and S333A single mutant strains, showing that only S333A abolishes the upper phosphorylated band. (D) Eight 2-fold serial dilutions of wild-type and ΔPrkC strains, confirming PrkC dependence of FtsZ phosphorylation.
(EPS)

**S3 Fig. GTPase assay analysis.** Raw GTPase activity data showing phosphate production rate versus FtsZ concentration for wild-type (A), S333A (B), and S333E (C) FtsZ. Solid lines represent individual replicates; each replicate was fit by linear regression, and the fit parameters were averaged to generate the final model (dashed lines). Slopes correspond to specific activities (GTP/min/FtsZ); x-intercepts correspond to critical concentrations (μM). (D) Overlay of the fitted models from A-C: wild-type (blue), S333A (orange), S333E (green).
(EPS)

**S4 Fig. Strain construction strategy for FtsZ expression.** (A) Organization of the native *ftsAZ* locus, showing the *ftsA-ftsZ* operon under control of its native promoter (P*az*). P*az* extends 371 bp upstream of the *ftsA* start codon into the *sbp*

gene and contains three SigA binding sites (P1, P2, and P3, orange). (B) The complete *ftsAZ* operon including its native P*az* promoter was integrated at *amyE* to achieve constitutive expression at native levels. (C) The native *ftsAZ* operon was replaced with a spectinomycin resistance cassette to eliminate wild-type FtsZ expression. (D) Western blot (top) and Ponceau stain (bottom) of wild-type, S333A, and S333E FtsZ harvested from strains (bWM200, bWM236, bWM237, respectively) grown in S7 minimal medium (left) and LB rich medium (right).
(EPS)

**S5 Fig. Assignment of tail residues through spectral overlap with synthetic peptide.** (A) $^1$H-$^{15}$N 2D-HSQC spectra overlay of the 15aa synthetic peptide (residues 328–342, blue) with the 67aa FtsZ tail (residues 316–382) in the absence (gray) and presence (orange) of unlabeled core. Peak overlap between the peptide and full-length tail enabled assignment of eight residues in the tail spectrum: L330, Q332, S333, I334, K335, H337/N338, and V341. These assignments revealed which tail residues are involved in core binding, as shown in Fig. 2C. (B) Quantification of peak intensity changes for assigned tail residues upon core binding. Log$_2$ fold change was calculated from peak heights normalized to median intensities. Positive values indicate intensity increases upon binding; negative values indicate decreases. The S333-I334-K335 motif shows the largest changes, while flanking residues show more modest changes.
(EPS)

**S6 Fig. AlphaFold predicts the S333 region contacts the core.** (A) Predicted aligned error (PAE) matrix from AlphaFold-Multimer upon co-folding of FtsZ core (residues 1–315) and CTL (residues 316–365). Lower PAE scores indicate higher confidence in predicted residue-residue distances. While PAE scores for the interface (12 Å at minimum) reflect the expected flexibility of interactions between a folded domain and an IDR, the same specific contacts are consistently predicted across multiple AlphaFold runs. (B) Minimum PAE scores between each CTL residue and the core. Within the 50aa CTL, AlphaFold specifically identifies residues N331-N338 as having the highest confidence contacts with the core (PAE < 15 Å), with the lowest scores (~12 Å) at S333, I334, and K335. These computationally predicted contact residues correspond precisely to the binding interface identified experimentally by NMR.
(EPS)

**S7 Fig. ITC identifies key binding residues on the tail and core.** (A) Representative ITC data for I334E (magenta) and K335A/H337A (blue) tail mutants titrated into wild-type FtsZ core. Neither showed detectable binding ($K_d$ > 2 µM), confirming these residues are essential for the tail-core interaction. (B-C) Representative ITC data for wild-type FtsZ tail titrated into M292H/I293W/F294K (B) and D210A/D213A (C) FtsZ core mutants. The MIF pocket triple mutant (B) displays aberrant binding with a biphasic isotherm, confirming this hydrophobic pocket is essential for normal tail-core interaction. The acidic patch mutant (C) shows approximately two-fold weaker binding ($K_d$ = 501 ± 64 nM) compared to wild-type core ($K_d$ = 228 ± 24 nM).
(EPS)

**S8 Fig. NOESY contact maps for S333 variant peptides.** Contact maps visualizing all ~120 manually assigned off-diagonal $^1$H-$^1$H NOE cross-peaks for each 15aa peptide variant: WT (A), S333A (B), S333E (C), pS333 (D). Intensities displayed on red scale (normalized peak heights, saturated at 5,000); irrelevant position-333 variants grayed out; black lines separate residues. Diagonal auto-peaks excluded. Complete NOE assignments provided in Table S6. Representative cross-peaks highlighted in Fig. 3C-E demonstrate progressive I334-K335 compaction across variants.
(EPS)

**S9 Fig. S333 modifications induce conformational restriction of the K335 sidechain.** Cross-sections through 2D $^1$H-$^1$H NOESY spectra at the K335 backbone amide frequency. Traces are normalized to wild-type maximum intensity. Vertical dashed lines mark the Hβ2 and Hβ3 peak positions in the wild-type spectrum. The characteristic doublet in WT

progressively coalesces through the S333A, S333E, and pS333 variants, with Hβ3 shifting toward Hβ2 while their intensity ratio equilibrates, ultimately merging into a single broad peak in pS333. These spectral changes indicate increasing conformational restriction of the K335 sidechain.
(EPS)

**S10 Fig. S333 modifications induce local structural changes in the tail's binding region.** (A) Overlay of $^1$H-$^{15}$N 2D-HSQC spectra of wild-type (black), S333A (blue), S333E (orange), and phospho-S333 (red) 15aa synthetic peptides (P328-P342). Circles demarcate groups of peaks that correspond to the same residue across all four variants. Peaks at position 333 are shown with dashed contours at reduced opacity. (B) Chemical shift perturbations (CSPs) for S333A, S333E, and phospho-S333 peptides relative to wild-type. CSPs were calculated as $\sqrt{[(\Delta\delta H)^2 + (\Delta\delta N/5)^2]}$ with values saturated at 0.10 ppm for visualization. Residues within the L330-H337 binding region show the largest positional changes. Position 333 is shown with gray hatching. (C) Peak intensity changes ($\log_2$ fold-change) for variant peptides relative to wild-type, normalized to reference residues S329, S340, and V341. Positive values indicate peak sharpening/stabilization; negative values indicate peak broadening/destabilization. Values are saturated at ±0.5 for visualization. Position 333 is shown with gray hatching.
(EPS)

**S11 Fig. Variant-specific NOE patterns distinguish S333 variants.** (A) Wild-type peptide NOESY strips showing Q332 and S333 HN spin systems. Q332 HN shows NOEs to its own Hβ2, Hβ3, and weak Hγ2. S333 HN shows NOEs to Q332 Hβ2, with modest Q332 Hβ3 contact and no Q332 Hγ2 interaction. (B) S333A peptide NOESY strips showing Q332, A333, and I334 HN spin systems. New NOEs appear between Q332 sidechains (Hβ2/Hβ3/Hγ2) and A333 HN, as well as a strong NOE between A333 Hβ and I334 HN, indicating formation of hydrophobic contacts between these three residues. (C) Phospho-S333 peptide NOESY strips showing Q332 and pS333 HN spin systems. pS333 HN displays a NOE pattern with strong Q332 Hβ2, moderate Q332 Hβ3, and weak Q332 Hγ2 contacts. This pattern resembles the Q332 interactions observed in S333A (panel B), suggesting phosphorylation maintains similar Q332 sidechain positioning while achieving I334-K335 compaction through a distinct mechanism.
(EPS)

**S12 Fig. MinC$_N$ and the FtsZ tail target the same surface of the FtsZ core.** AlphaFold models of the FtsZ core bound to MinC$_N$ (left, adapted from Machado et al. 2025 [35]) or the FtsZ CTL (right). Both fragments are predicted to bind the core's C-terminal polymerization surface. Residues involved in binding are colored as in Fig 4B (left) and Fig 3A (right).
(EPS)

**S13 Fig. FtsZ tail mutations restore MinC$_N$ access to full-length FtsZ.** Representative ITC data of MinC$_N$ titrated into full-length FtsZ containing S333A (A), S333E (B), or K335A/H337A (C) tail mutations. S333A and S333E mutations restore normal MinC$_N$ binding with $K_d$ values comparable to isolated core. The K335A/H337A double mutant shows multiple MinC$_N$ molecules binding per FtsZ, indicating significantly altered engagement of MinC$_N$ with FtsZ. All three mutations demonstrate that tail-core binding normally occludes the MinC$_N$ site. (D) Representative ITC data showing the wild-type FtsZ tail cannot bind to full-length FtsZ ($K_d > 2$ μM), confirming that each core's binding site is already occupied by its own tethered tail. This intramolecular tail-core interaction (*in cis*) prevents binding of exogenous tails (*in trans*), consistent with the 1:1 stoichiometry observed for *trans* tail-core binding in Fig 2A.
(EPS)

**S1 Appendix. Supplementary Information.** Contains Tables A-F, detailed construction protocols, sequences, and supplementary references.
(DOCX)

**S1 Table. ITC binding parameters.** Isothermal titration calorimetry binding parameters for all replicates. "No Binding" indicates $K_d > 2\ \mu M$.
(XLSX)

**S2 Table. Tail binding HSQC peaks.** $^1H$-$^{15}N$ HSQC peak assignments for the 67aa FtsZ tail (residues 316–382) in free and core-bound states (1:1 molar ratio). Peak assignments were transferred from the wild-type 15aa synthetic peptide based on chemical shift overlap.
(XLSX)

**S3 Table. Tail binding HSQC analysis.** Chemical shift perturbations (CSPs) and normalized intensity changes ($\log_2$ fold-change) for assigned tail residues upon core binding.
(XLSX)

**S4 Table. Peptide HSQC peaks.** $^1H$-$^{15}N$ HSQC peak assignments for 15aa synthetic peptides (P328-P342) representing the FtsZ tail binding region. Includes wild-type, S333A, S333E, and phospho-S333 variants. Peak heights normalized to median.
(XLSX)

**S5 Table. Peptide HSQC analysis.** Chemical shift perturbations (CSPs) and normalized intensity changes ($\log_2$ fold-change) for each peptide variant relative to wild-type.
(XLSX)

**S6 Table. Peptide NOESY peaks.** $^1H$-$^1H$ NOESY peak assignments for all four peptide variants. Peak heights normalized to median.
(XLSX)

**S7 Table. NMR normalization parameters.** Median normalization factors and baseline contour thresholds used for HSQC and NOESY spectra of all peptide variants (see Methods).
(XLSX)

## Acknowledgments

This work was supported by funding from Harvard University. W.J.M. is deeply grateful to Shicong Xie for her invaluable mentorship, intellectual guidance, and extensive feedback throughout all stages of this research. Her insights and thoughtful discussions were instrumental in shaping this work. W.J.M. also thanks Alexandre Bisson-Filho for helpful discussions on FtsZ biochemistry.

## Author contributions

**Conceptualization:** William J Mallard, Vincent V. Pham.

**Data curation:** William J Mallard.

**Formal analysis:** William J Mallard, Vincent V. Pham.

**Investigation:** William J Mallard, Vincent V. Pham.

**Methodology:** William J Mallard, Vincent V. Pham.

**Project administration:** William J Mallard.

**Software:** William J Mallard.

**Validation:** William J Mallard, Vincent V. Pham.

**Visualization:** William J Mallard.

**Writing – original draft:** William J Mallard.

**Writing – review & editing:** William J Mallard, Vincent V. Pham.

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
