## [Decision Letter · Decision Letter 0]

28 Mar 2025

PONE-D-25-08162FtsZ phosphorylation modulates tail-core binding to tune cell division in Bacillus subtilisPLOS ONE

Dear Dr. Mallard,

Thank you for submitting your manuscript to PLOS ONE. After careful consideration, we feel that it has merit but does not fully meet PLOS ONE’s publication criteria as it currently stands. Therefore, we invite you to submit a revised version of the manuscript that addresses the points raised during the review process.

We look forward to receiving your revised manuscript.

Kind regards,

Hari S. Misra, Ph.D.

Academic Editor

PLOS ONE

Journal Requirements:

4. We note that this submission includes NMR spectroscopy data. We would recommend that you include the following information in your methods section or as Supporting Information files:

1) The make/source of the NMR instrument used in your study, as well as the magnetic field strength. For each individual experiment, please also list: the nucleus being measured; the sample concentration; the solvent in which the sample is dissolved and if solvent signal suppression was used; the reference standard and the temperature.

2) A list of the chemical shifts for all compounds characterised by NMR spectroscopy, specifying, where relevant: the chemical shift (δ), the multiplicity and the coupling constants (in Hz), for the appropriate nuclei used for assignment.

3)The full integrated NMR spectrum, clearly labelled with the compound name and chemical structure.

We also strongly encourage authors to provide primary NMR data files, in particular for new compounds which have not been characterised in the existing literature. Authors should provide the acquisition data, FID files and processing parameters for each experiment, clearly labelled with the compound name and identifier, as well as a structure file for each provided dataset. See our list of recommended repositories here: https://journals.plos.org/plosone/s/recommended-repositories

Reviewers' comments:

Reviewer's Responses to Questions

**Comments to the Author**

1. Is the manuscript technically sound, and do the data support the conclusions?

Reviewer #1: Yes

Reviewer #2: Partly

2. Has the statistical analysis been performed appropriately and rigorously? 

Reviewer #1: Yes

Reviewer #2: Yes

3. Have the authors made all data underlying the findings in their manuscript fully available?

Reviewer #1: Yes

Reviewer #2: Yes

4. Is the manuscript presented in an intelligible fashion and written in standard English?

Reviewer #1: Yes

Reviewer #2: Yes

5. Review Comments to the Author

Reviewer #1: Comments to the authors:

This manuscript by Mallard and Pham explores an interesting aspect of regulation of FtsZ by phospohorylation. The authors identify S333 residue within the CTL to be phosphorylated in a PrkC kinase dependent manner. They further show that the FtsZ mutants S333A and S333E have enhanced GTPase activity and a lower critical concentration of polymerisation. Accordingly, cell size is also shown to be smaller than WT upon expression of these mutant FtsZs. Alphafold structure predictions and interactions also support the idea that the CTL binds the core region engaging helix H10 and β-strand S9. The CTL peptide is also shown to interact with FtsZ core region by ITC and NMR experiements. Interestingly, the CTL can effectively compete with MinCN for interaction with FtsZ.

The findings reported are interesting and suggest a regulatory role for FtsZ phosphorylation in cell cycle in Bacillus subtilis. The conclusions drawn are generally well supported by the experiments and data. While, in support of the article being suitable for publication, I have certain queries.

1.I’m intrigued by the model proposed in Figure 4. The authors mention that when the tail is phosphorylated, intramolecular tail-core interaction is disrupted. Although the authors provide a reasonable structural explanation for the similar defects seen with both S333A and S333E mutations, I think these conclusions may be strengthened by testing the binding of the core residue mutants K335A/H337A to MinCN. One would expect that these mutants are not competed out by the CTL and would these FtsZ mutants be more sensitive to inhibition by MinCN for Z-ring inhibition.

2.Similarly, do S333A and S333E CTL mutants fail to compete with MinCN binding.

3.How does the GTPase activity of FL-FtsZ compare when extrinsic S333A/ E CTL is added? Would it possible to obtain a CTL with Phosphoserine at 333 and compare the differences in the GTPase activity and critical concentration of polymerisation when extrinsically added to FL-FtsZ?

I feel, these few in vitro experiments can really add value to the impact of the manuscript before being accepted for publication.

4.Also, although not essential, does depletion of any of the bundlers of FtsZ in S333A and S333E result in restoration of cell length to WT-like? If this not possible to test experimentally, some thoughts of the authors along division ring stabilisation by phosphor regulation could be included in the discussion.

Reviewer #2: In this article, the authors convincingly show that the essential cell division protein conserved in nearly all bacteria, FtsZ, is able to autoregulate its enzymatic activity in the Gram-positive model bacterium Bacillus subtilis. This is achieved via a key serine residue (S333) in the intrinsically disordered region which is subject to phosphorylation by a Ser/Thr kinase PrkC. Furthermore, the authors reveal that the disordered C-terminal tail (CTT) is able to compete with a FtsZ regulator (MinC) for the same interaction surface in the FtsZ’s enzymatic core region. Overall, this report provides a valuable insight into the specific role phosphorylation plays in regulating FtsZ in B. subtilis. Moreover, this finding is broadly influential as it provides a plausible explanation regarding the significance of FtsZ phosphorylation reported in multiple species. However, I do have the following concerns which moderately dampened my enthusiasm.

Major Comments:

1. Fig. 1E: What is the WT used? Is it bWM200 or PY79? In the text they describe native ftsAZ promoter, is it the strain shown in S5B or S5D? If it is IPTG-inducible ftsAZ, then what concentration was used for expression? Could the authors show a western blot to confirm all three (full-length FtsZ, S333A, S333E) are all produced at similar levels? This is important as slight variations in FtsZ level affect cell length as the authors indicate in the text; and this is the only physiologically relevant data presented in this report. Indicating the strain numbers in the figure legend would be helpful for replicating these results in the future. Is this reproducible in another growth medium such as LB - if not, why? If MinC has unlimited access to FtsZ in S333A/E mutants, then shouldn’t we expect longer cells?

2. ITC experiments: The dissociation constant between Core-CTT and Core-MinC interactions in the nanomolar range seems atypical. Most FtsZ partners are usually in the micromolar range (PMIDs: 22473211; 30636070). Although the following experiment is not needed, a better competition would be with FtsZ core + MinC, and titrating CTT to help support the claim “Surprisingly, MinCN binding was completely abolished when titrated into full-length FtsZ (Fig. 3E, F). This is consistent with the CTT’s higher affinity for the core (Kd = 66 nM vs 111 nM for MinCN).”

3. In the last sentence of Results section, the authors indicate “these results demonstrate direct competition between the tail’s S333 region and MinCN for the C-terminal polymerization surface.” Although plausible, unless the authors make specific mutations in the core to disrupt both MinC and CTT binding, it cannot be argued that this competition is direct. It could possibly be allosteric regulation.

4. In the Discussion: “For filaments, our GTPase assays show S333 mutations affect catalysis, suggesting the tail still binds the core and thus may protect GDP-bound FtsZ interfaces from MinCN, though further structural studies will be needed to confirm this mechanism.” Why is this the case? If S333 mutants do not inhibit the core, higher GTPase activity is expected. The results from this publication may support the notion presented here: PMID: 32198113 (and should be cited).

5. “Whereas its homologs phosphorylate FtsZ in other bacteria (Supp. Table 1), such a direct link to division has not been shown in B. subtilis.” Phosphoproteomics data is experimental evidence, as such it is a direct link. Although, I agree in B. subtilis the explicit connection has not been made. As a side note, it appears that residue T323 is also phosphorylatable (PMID: 34473931).

Minor Comments:

1.Line numbering would be useful for making comments.

2.Supplementary figure callouts are not sequential in the text.

3.prkC gene name vs PrkC protein name - standard nomenclature is not being followed. For example, in Fig. 1B it should be ΔprkC.

4.For the phostag gel experiments, I don’t see Halo-S333E in the strain list.

5.NMR: Is there data for phosphomutants to show the chemical shift changes are not observed with a S333 mutant?

6.Selection of H337A needs some introduction.

7.Could helix H10 and beta strand S9 be labeled in the figure?

8.Could the authors discuss the mechanism and/or interaction surface used by MciZ (PMIDs: 25848052; 30636070)? This could add more value to their study if this CTL-core interaction is a general mechanism for competing with negative regulators. Also, how does interaction partners that bind to CTP affect the significance/relevance of phosphorylation (PMID: 23457247)? A brief discussion on this could also be beneficial.

9.Is it possible FtsZ CTT perform intermolecular regulation perhaps in laterally associated protofilaments (in addition to intramolecular as proposed), PMID: 29089389?

6. PLOS authors have the option to publish the peer review history of their article (what does this mean? ). If published, this will include your full peer review and any attached files.

**Do you want your identity to be public for this peer review?** For information about this choice, including consent withdrawal, please see our Privacy Policy .

Reviewer #1: No

Reviewer #2: No

---

## [Author Response · Author response to Decision Letter 1]

27 Oct 2025

We thank both reviewers for their constructive comments, which have substantially strengthened our manuscript. Their suggestions prompted us to perform additional experiments that revealed the mechanistic basis for our initial observations: FtsZ’s tail directly binds its core through an interaction centered on S333, and phosphorylation disrupts this binding. In response to their feedback, we have:

1. Performed extensive new experiments including competitive binding assays between MinC and tail mutants (Reviewer 1), displacement experiments showing tail dominance over MinC (Reviewer 2), ITC analysis of core mutants to demonstrate direct competition rather than allostery (Reviewer 2), cell length measurements in rich media revealing context-dependent phenotypes (Reviewer 2), Western blots confirming comparable FtsZ expression levels (Reviewer 2), and NMR structural analysis of authentic phospho-S333 peptides.

2. Significantly expanded the manuscript with new Results sections describing tail-core binding and its structural basis (lines 244-312), expanded Discussion of regulatory mechanisms (lines 388-413), and many new supplementary figures and tables providing comprehensive supporting data.

3. Addressed all technical concerns including correcting nomenclature, adding line numbers, ensuring sequential figure citations, and including strain identifiers in all figure legends.

These revisions have transformed our initial phosphorylation study into a mechanistic analysis of how S333 serves as a phospho-switch that modulates tail-core binding, thereby affecting both FtsZ polymerization dynamics and regulatory protein access. We address each specific comment in detail below.

Response to Reviewer #1

We thank Reviewer #1 for their positive assessment of our work and for acknowledging that our findings “suggest a regulatory role for FtsZ phosphorylation in cell cycle in Bacillus subtilis” and that “the conclusions drawn are generally well supported by the experiments and data.”

Comment 1: "I'm intrigued by the model proposed in Figure 4. The authors mention that when the tail is phosphorylated, intramolecular tail-core binding is disrupted. Although the authors provide a reasonable structural explanation for the similar defects seen with both S333A and S333E mutations, I think these conclusions may be strengthened by testing the binding of the core residue mutants K335A/H337A to MinCN. One would expect that these mutants are not competed out by the CTL and would these FtsZ mutants be more sensitive to inhibition by MinCN for Z-ring inhibition."

Comment 2: "Similarly, do S333A and S333E CTL mutants fail to compete with MinCN binding."

Response to Comments 1 and 2:

Done. See Figures S12B-D and Results lines 333-342.

We thank the reviewer for these excellent suggestions. We have performed the suggested competition assays by titrating MinCN into full-length FtsZ with tail mutations (S333A, S333E, and K335A/H337A). As predicted, MinCN bound all three tail mutants (Fig. S12B-D), whereas it did not bind wild-type full-length FtsZ (Fig. 4C). These results confirm that disrupting the tail-core interaction (either through mutations at S333 or at K335/H337) exposes the MinCN binding site, providing strong support for our model of competitive binding between the CTL and MinCN. We have incorporated these results into the revised manuscript.

Comment 3: "How does the GTPase activity of FL-FtsZ compare when extrinsic S333A/E CTL is added? Would it possible to obtain a CTL with Phosphoserine at 333 and compare the differences in the GTPase activity and critical concentration of polymerisation when extrinsically added to FL-FtsZ? I feel, these few in vitro experiments can really add value to the impact of the manuscript before being accepted for publication."

Response to Comment 3:

We have addressed this through new ITC experiments (Figure S12E, Results lines 339-342) and new NMR experiments (Figure 3C-F, S8-11, Tables S10-13, Results lines 244-312).

We appreciate the reviewer’s suggestion and have performed additional ITC experiments that directly address this question. Our data demonstrate that exogenous tail cannot affect full-length FtsZ activity for the following reasons:

First, wild-type tail shows no detectable binding to full-length FtsZ (Kd > 2 μM; new Figure S12E) because each FtsZ molecule’s binding site is already occupied by its own intramolecular tail. This cis occupation prevents any trans interaction with exogenous tails, wild-type or otherwise.

Second, S333A and S333E tail mutants completely abolish binding even to the isolated FtsZ core (Kd > 2 μM; Figure 3B), despite the core lacking a competing intramolecular tail. Since these mutants cannot bind even an available site, they certainly cannot compete with the higher-affinity wild-type tail already occupying each site in full-length FtsZ.

Therefore, adding exogenous S333A/E tails would have no effect on FtsZ GTPase activity, as they cannot engage the binding site necessary to influence polymerization. Our ITC experiments provide more mechanistic insight than GTPase assays would, as they directly demonstrate the absence of the binding interaction required for any functional effect.

Regarding phosphorylated tail: While we successfully synthesized a 15aa phospho-peptide for our NMR studies, the full 67aa tail exceeds practical synthesis limits (~30aa). In vitro phosphorylation would require recapitulating the activation of PrkC—an RD kinase whose activation depends on its extracellular PASTA domains interacting with Lipid II in the membrane to drive dimerization and activation of the cytoplasmic kinase domain—which is not feasible for this study. Expression methods that directly incorporate phosphoserine (such as amber codon suppression) would require specialized genetic systems and typically yield insufficient protein for biochemical assays.

However, our new NMR data with a 15aa phospho-S333 peptide (Figure 3C-F, S8-11, Tables S10-13, Results lines 244-312) demonstrates that phosphorylation of S333 induces the same structural compaction of I334-K335 observed with S333E, validating our use of S333E as a phospho-mimetic throughout this study.

Comment 4: "Also, although not essential, does depletion of any of the bundlers of FtsZ in S333A and S333E result in restoration of cell length to WT-like? If this not possible to test experimentally, some thoughts of the authors along division ring stabilisation by phosphor regulation could be included in the discussion."

Response to Comment 4:

Addressed conceptually below.

We thank the reviewer for this insightful suggestion. In B. subtilis, the primary FtsZ filament bundlers are SepF and ZapA. SepF is membrane-associated and interacts with FtsZ's C-terminal peptide (CTP), while ZapA forms tetramers that interact with FtsZ’s core, though the binding interface is unknown.

Bundler deletions are known to make cells longer. For example, ΔzapA reduces FtsZ filament bundling, which impairs Z-ring condensation, which delays division, which makes cells longer. The reviewer raises an interesting question: could the increased FtsZ polymer abundance in S333 mutants (due to their lower critical concentration) compensate for reduced bundling activity in a double mutant, potentially restoring wild-type cell length?

While this is an intriguing hypothesis, testing it properly would be technically challenging. Both SepF and ZapA are encoded in multi-gene operons, so creating clean depletion strains without affecting other co-regulated genes is complex. Additionally, the relationship between bundler levels and cell length is likely non-linear, and bundler depletion could also have secondary effects on cell physiology that would be difficult to disentangle. Given these considerations and time constraints, this investigation extends beyond the scope of the current study.

However, we considered whether tail-core binding could affect bundling. After careful analysis, we concluded that tail phosphorylation likely does not directly modulate bundler function because: (1) FtsZ must polymerize before bundlers can act, and (2) once polymerized, the tail cannot rebind its core since the C-terminal surface is occupied by the next monomer. Therefore, bundlers interact with FtsZ filaments whose tails are already unable to bind their cores, regardless of phosphorylation state.

Any effect of S333 mutations on bundling would be indirect: enhanced polymerization provides more substrate for bundling proteins. Whether this increased polymer abundance compensates for or exacerbates bundling defects remains an interesting question for future investigation. We have revised our Discussion to acknowledge that S333 mutations alter polymerization dynamics, which could indirectly affect Z-ring organization through changes in the available bundling substrate.

Response to Reviewer #2

We thank Reviewer #2 for their positive assessment that our report “provides a valuable insight into the specific role phosphorylation plays in regulating FtsZ in B. subtilis” and “is broadly influential as it provides a plausible explanation regarding the significance of FtsZ phosphorylation reported in multiple species.”

Major Comments

Comment 1a: "Fig. 1E: What is the WT used? Is it bWM200 or PY79? In the text they describe native ftsAZ promoter, is it the strain shown in S5B or S5D? If it is IPTG-inducible ftsAZ, then what concentration was used for expression?"

Response to Comment 1a:

Done. Clarified throughout the manuscript.

We apologize for the lack of clarity regarding strains. For Fig. 1E, all strains express the ftsAZ operon from an ectopic locus (amyE) under the native ftsAZ promoter (Paz) as shown in Fig. S4B, with the native ftsAZ operon knocked out as shown in Fig. S4C. The WT, S333A, and S333E ftsZ variants correspond to strains bWM200, bWM236, and bWM237, respectively. We have clarified this throughout the manuscript. The Phyperspank construct was a remnant from an earlier draft, and we have now removed it to avoid confusion.

Comment 1b: "Could the authors show a western blot to confirm all three (full-length FtsZ, S333A, S333E) are all produced at similar levels? This is important as slight variations in FtsZ level affect cell length as the authors indicate in the text; and this is the only physiologically relevant data presented in this report."

Response to Comment 1b:

Done. See new Figure S4D.

We performed Western blots to compare FtsZ protein levels across wild-type and mutant strains. FtsZ levels are comparable in all three strains, with variations of less than 10% relative to the wild-type control. Given that a two-fold increase in FtsZ protein is reported to reduce cell length by only 10% (Weart and Levin, 2003), the small variations we observe in protein levels (which are within the margin of error for quantitative Western blots) cannot account for the 5-7% decrease in cell length observed with the S333 mutants. This confirms that the cell length phenotype is due to the specific effects of the S333 mutations on FtsZ function rather than differences in expression levels. We have included these Western blot results as a new panel in Fig. S4D.

Comment 1c: "Indicating the strain numbers in the figure legend would be helpful for replicating these results in the future."

Response to Comment 1c:

Done. Updated all figure legends.

We have now included strain numbers in all figure legends to facilitate replication of our results.

Comment 1d: "Is this reproducible in another growth medium such as LB - if not, why?"

Response to Comment 1d:

Done. Tested in CH rich medium (Figure 1F, Results lines 134-140, Discussion lines 402-413).

As suggested, we tested whether the cell length phenotype is reproducible in rich media. LB medium is highly autofluorescent, making it incompatible with fluorescent imaging of stained membranes that is necessary to resolve individual cells within chains. We therefore used CH medium, another rich medium commonly used for B. subtilis.

Interestingly, while S333 mutants produce shorter cells in minimal media (5-7% reduction), they result in longer cells in rich media (S333A: 5% increase, S333E: 7% increase). We believe the most important observation here is that S333 mutations significantly impact cell size in both media conditions, confirming this residue’s fundamental involvement in cell size regulation.

Notably, in both conditions, the S333E mutation consistently produces a stronger effect than S333A, with S333A falling between wild-type and S333E. This gradient of phenotypic severity (where S333E consistently shows stronger effects than S333A) parallels the gradient of structural changes we observed via NMR (where WT < S333A < S333E < pS333). This further supports our use of S333E as a phospho-mimetic.

These context-dependent effects provide additional insights into S333’s role in integrating multiple regulatory inputs to FtsZ. This nutrient-dependent reversal of phenotype is consistent with previous work by Chien et al. (2012), who demonstrated that cell size in B. subtilis is regulated by metabolic sensors like UgtP, which inhibits FtsZ assembly specifically in nutrient-rich conditions. In minimal media, UgtP is sequestered away from FtsZ, while in rich media, UgtP actively interacts with FtsZ to delay division and increase cell size. However, the molecular interface between UgtP and FtsZ has not been experimentally determined.

To further explore the molecular basis for these context-dependent effects, we used AlphaFold Multimer to predict potential interaction sites between UgtP and FtsZ. Intriguingly, AlphaFold predicts that UgtP binds to FtsZ’s C-terminal polymerization surface — the same region where we have demonstrated FtsZ’s tail binds. This is not surprising, as polymerization surfaces are a logical regulatory target for any polymer system, and FtsZ’s C-terminal polymerization surface in particular is known to interact with two other division regulators (MinC, MciZ). This AlphaFold prediction suggests a model where the FtsZ tail and UgtP may compete for the same binding site on the FtsZ core.

In this competitive binding model:

1. In minimal media (low UgtP activity): S333 mutations disrupt tail-core binding, increasing the availability of the C-terminal polymerization surface for FtsZ-FtsZ interactions. This enhances FtsZ assembly and results in shorter cells.

2. In rich media (high UgtP activity): S333 mutations disrupt tail-core binding, potentially increasing UgtP’s access to FtsZ molecules. This might impair filament bundling or stability, resulting in longer cells.

This model suggests that S333 phosphorylation may modulate FtsZ's responsiveness to nutrient-dependent regulators like UgtP, providing a mechanism for integrating multiple cellular signals to fine-tune cell division.

Comment 1e: "If MinC has unlimited access to FtsZ in S333A/E mutants, then shouldn't we expect longer cells?"

Response to Comment 1e:

Mechanistic explanation provided here.

This is an excellent question that touches on the complexity of FtsZ regulation. Our data show that in S333A/E mutants, MinCN indeed has increased access to FtsZ’s C-terminal polymerization surface, as demonstrated by our competitive binding experiments. Intuitively, one might expect this to result in longer cells, as MinC is a negative regulator of FtsZ polymerization. However, there are several key factors to consider:

1. Competing effects on FtsZ polymerization: S333 mutations have two opposing effects on FtsZ: they increase MinC accessibility (which would inhibit polymerization) but also significantly lower the critical concentration for FtsZ assembly (by up to 27%) as shown in Fig. 1C. Our cell length data suggests that in minimal media, the enhanced intrinsic polymerization properties of the mutants dominate over increased MinC accessibility.

2. Physiological context of MinC activity: MinC’s inhibitory function is spatially regulated. In B. subtilis, MinC is largely confined to cell poles through the MinCDJ-DivIVA system, with minimal activity at mid-cell where the Z-ring forms. Therefore, increased MinCN binding to FtsZ may not translate directly to division inhibition

---

## [Editor Report · Decision Letter 1]

13 Nov 2025

FtsZ phosphorylation modulates tail-core binding to tune cell division in Bacillus subtilis

PONE-D-25-08162R1

Dear Dr. Mallard,

We’re pleased to inform you that your manuscript has been judged scientifically suitable for publication and will be formally accepted for publication once it meets all outstanding technical requirements.

Kind regards,

Hari S. Misra, Ph.D.

Academic Editor

PLOS ONE
---

## [Editor Report · Acceptance letter]

PONE-D-25-08162R1

PLOS One

Dear Dr. Mallard,

I'm pleased to inform you that your manuscript has been deemed suitable for publication in PLOS One. Congratulations! Your manuscript is now being handed over to our production team.

Kind regards,

on behalf of

Professor Hari S. Misra

Academic Editor

PLOS One